# The eukaryotic-like characteristics of small GTPase, roadblock and TRAPPC3 proteins from Asgard archaea
Linh T. Tran[1,4], Caner Akıl[1,2,4], Yosuke Senju [1] & Robert C. Robinson [1,3] ✉

Membrane-enclosed organelles are defining features of eukaryotes in distinguishing these organisms from prokaryotes. Specification of distinct membranes is critical to assemble and maintain discrete compartments. Small GTPases and their regulators are the signaling molecules that drive membrane-modifying machineries to the desired location. These signaling molecules include Rab and Rag GTPases, roadblock and longin domain proteins, and TRAPPC3-like proteins. Here, we take a structural approach to assess the relatedness of these eukaryotic-like proteins in Asgard archaea, the closest known prokaryotic relatives to eukaryotes. We find that the Asgard archaea GTPase core domains closely resemble eukaryotic Rabs and Rags. Asgard archaea roadblock, longin and TRAPPC3 domain-containing proteins form dimers similar to those found in the eukaryotic TRAPP and Ragulator complexes. We conclude that the emergence of these protein architectures predated eukaryogenesis, however further adaptations occurred in proto-eukaryotes to allow these proteins to regulate distinct internal membranes.

One of the defining characteristics of eukaryotic cells is the presence of complex intracellular membranes. By contrast, bacteria and archaea are generally smaller and devoid of such internal membranes. Eukaryotes have complex systems for assembling, maintaining, and modifying these various membrane partitions. Many of these processes are orchestrated by small GTPases of the RAS superfamily, which recruit the various protein machines to specific membranes[1]. Membrane trafficking, the transport of components between discrete compartments, proceeds through the budding of vesicles from a donor compartment and the fusion of vesicles to the acceptor compartment under the direction of small GTPases. Precision in vesicle trafficking is essential to maintain and communicate between distinct cellular compartments, such as endosomes, lysosomes, Golgi apparatus, endoplasmic reticulum, and the plasma membrane. Thus, one of the major adaptations during eukaryogenesis was the evolution of specific membrane machineries responsible for targeted vesicle traffic.

In eukaryotic vesicle trafficking, Arf GTPase complexes control the budding of vesicles, whilst Rab GTPase complexes are the regulators that specify the sort codes to target vesicles to acceptor membranes[2]. Eukaryotic Rabs are high-affinity GTP and GDP-binding proteins with low-level GTPase activity that operate as binary molecular switches[3]. The inactive GDP-bound Rabs are recruited to the acceptor membrane by guanine nucleotide exchange factors (GEFs) that exchange the Rab-bound nucleotide to GTP[4]. The active GTP and membrane-bound Rabs recruit or activate effector proteins leading to events such as the fusion of vesicles. GTP hydrolysis in Rabs, often stimulated by GTPase-activating proteins (GAPs), leads to the dissociation of the Rabs and effectors from the acceptor membrane. A second element in membrane targeting of Rabs is the geranylgeranylation of C-terminal cysteine residues[3]. Nascent Rabs are bound by Rab escort protein (REP), which presents the Rab C-terminus to a Rab geranylgeranyl transferase protein for geranylgeranylation. The modified Rab is shuttled to and from the target membrane as soluble complexes with REP or GDP dissociation inhibitor (GDI), where the geranylgeranyl modification anchors the Rab in the target membrane.

GEFs are structurally diverse, and can form large complexes which often contain subunits of the roadblock (RB)/longin families of proteins[5]. These are dimeric platforms that often interact directly with small GTPases and act mainly as GEFs, and occasionally as GAPs, though allosteric mechanisms or through assembling additional components[5–8]. The orientation of small GTPase proteins in binding to RB/longin dimers is conserved in evolution from prokaryotes to eukaryotes[5]. Since RB proteins are abundant in archaea and have a sporadic distribution in bacteria, RB/longin families of proteins are thought to be of archaeal origin[9]. In the bacterium

[1]Research Institute for Interdisciplinary Science, Okayama University, Okayama 700-8530, Japan. [2]Division of Structural Biology, University of Oxford, Oxford, England. [3]School of Biomolecular Science and Engineering (BSE), Vidyasirimedhi Institute of Science and Technology (VISTEC), Rayong 21210, Thailand. [4]These authors contributed equally: Linh T. Tran, Caner Akıl. ✉e-mail: robert.b@vistec.ac.th

*Myxococcus xanthus*, the mutual gliding motility A protein (MglA, a small GTPase) binds to the GEF/GAP protein MglB, a RB homodimer. The dual GEF/GAP activities of MglB, cycle the nucleotide state in MglA, and function to oscillate cell polarity in this bacterium[10]. The eukaryotic TRAnsport Protein Particle (TRAPP) complexes (TRAPP II and III) act as GEFs for Rab1 or Rab11 (Ypt1p and Ypt31p/Ypt32p in yeast, respectively)[11,12]. TRAPP II controls Golgi traffic, while TRAPP III regulates post-Golgi traffic and autophagy[13]. At the core of these TRAPP complexes lies the longin heterodimer (TRAPPC1/C4 in mammals and Bet5p/Trs23p in yeast) flanked by TRAPP domain subunits (TRAPPC3/C5 in mammals and Bet3p/Trs31p in yeast)[12,14]. In yeast, the longin heterodimer (Bet5p/Trs23p) interacts with Ypt1p (Rab1) in a geometry similar to the MglA/B complex[5].

Some RB/longin domain proteins have roles that are not involved in small GTPase regulation, for instance longin domains are found in SNARE proteins, however these other functions often involve membrane modulation[15]. A second system is the Ragulator-Rag complex which is involved in localizing the TORC1 metabolic sensing complex on the lysosome[16,17]. The Ragulator complex comprises four RB domains that form two pairs of heterodimers (LAMTOR2/3 and LAMTOR4/5) which are held together at the lysosome by the extended protein LAMTOR1[18]. LAMTOR4/5 is comprised of truncated RB subunits lacking the C-terminal helix, which we refer to as $RB_{LC7}$ due to the existence of this architecture in dynein light chain[9]. The Ragulator complex acts as a scaffold to recruit heterodimer Rag GTPases (A/C, A/D, B/C, or B/D), and the longin domain-containing Foliculin/Foliculin interacting protein (FLCN:FNIP) complex, to the lysosome[19]. Nutrient stimulation stimulates RagA and the FLCN:FNIP complex to promote an exchange of the nucleotides from $^{GDP}$RagA:RagC$^{GTP}$ to $^{GTP}$RagA:RagC$^{GDP}$, to engage TORC1 in regulating lysosome activity, biogenesis, and positioning[16,19–21].

Metagenomic sequencing of environmental sediment samples has identified a superphylum of archaea, Asgard archaea (Asgard), which contain genes that were previously thought to be exclusive to eukaryotes[22–24]. Protein sequences displaying homology to the TRAPP complex and Ragulator complex subunits are encoded in the genomes of this superphylum. Lokiarchaeota (Loki), Odinarchaeota (Odin), and Thorarchaeota (Thor) possess potential Rab and Rag GTPases, and RB homologs, whilst longin homologs are found in Loki and Odin, and TRAPPC3 (Bet3) homologs are found in Thor, with more distant superfamily V4R proteins in Loki[2,25]. In Loki, 37 Rab-like proteins, 38 RB domains and 41 longin domains have been predicted[2,26]. However, none of these homologs has been analyzed at the protein level. In the two Asgard archaea to be isolated, *Candidatus Prometheoarchaeum syntrophicum* strain MK-D1 (MKD1) and *Candidatus Lokiarchaeum ossiferum* (Loki ossiferum) which are both Lokis, no internal membrane compartments have been observed[27,28]. By contrast, extracellular vesicles and membrane extensions are plentiful.

Since Asgard archaea are thought to share a common ancestor with eukaryotes, analysis of their TRAPP and Ragulator complex subunits at the protein level is an important step in assessing the pre-eukaryotic emergence of vesicle trafficking machineries. Here, we adopted an X-ray crystallography approach to compare the structures of a potential Asgard Rab small GTPase, several RBs and two TRAPPC3 proteins to their eukaryotic counterparts, with a focus on Thor and MKD1. We found common structural features within each class of protein, suggesting that Asgard archaea are likely to possess sophisticated mechanisms of membrane regulation in line with their complex morphologies[27,28].

## Results

### Thor-Rab

To establish the authenticity of Asgard Rab-like small GTPases at the protein level, we selected a predicted small GTPase encoding gene (GenBank accession number KXH73347.1) from the *Candidatus Thorarchaeota* SMTZ1-45 archaeon MAG (GenBank accession number LRSL01000056.1) based on sequence considerations and solubility on heterologous expression in *E. coli*. The protein (Thor-Rab) was purified, and

crystallized in the presence of GTP and GDP (5 mM each). The crystals contain two molecules of Thor-Rab in the asymmetric unit. The resulting crystal structure, refined to 1.5 Å resolution, revealed that GDP/Mg$^{2+}$ was bound to both copies of Thor-Rab in the crystallographic asymmetric unit (Fig. 1a, Supplementary Figs. 1 and 2 and Supplementary Table 1). Subsequently, we co-crystallized Thor-Rab with the slow hydrolyzing GTP-γS in the absence of Mg$^{2+}$. The structure, refined 1.75 Å resolution, contains mixed occupancy for GDP/GTP-γS in one copy of Thor-Rab and the full occupancy for GTP-γS in the second copy (Supplementary Figs. 1 and 2 and Supplementary Table 1). Finally, a GTP-γS/Mg$^{2+}$ soak (5 mM each, 10 min) of a Thor-Rab/GDP crystal, refined at 1.95 Å resolution, contained mainly GTP-γS at both sites with only one site displaying robust density for Mg$^{2+}$ (Supplementary Figs. 1 and 2 and Supplementary Table 1). Molecule 1 in each crystal contains a partially ordered Switch I loop and a fully traceable Switch II loop in the GDP-bound structure (Supplementary Figs. 1 and 2g). Molecule 2 in the GDP-bound structure contains a mostly traceable Switch I loop and a partially ordered Switch II loop (Supplementary Figs. 1 and 2h). Thus, the conformations of the Switch I and Switch II loops did not change substantially in the presence of different nucleotides (Supplementary Fig. 1i, j). We interpret this to indicate that the conformational state of Thor-Rab is dominated by the crystal packing (Fig. 2), which holds it in a state that allows nucleotide exchange.

Thor-Rab contains the typical core domain that is found in eukaryotic small GTPase proteins. We further characterized the structural relatedness to other small GTPases via structural superimposition. Thor-Rab shows the highest structural homology to human Rab-1B in the current entries in the Protein Database (PDB), characterized by an RMSD of 1.04 Å over 159 aligned residues (Fig. 1b). The Thor-Rab structure is similar but more distant from prokaryotic MglA small GTPases, such as that from the Gram-negative bacterium *Thermus thermophilus* HB8 (2.38 Å over 144 residues, Fig. 1c). Since the structural relationships may be skewed by the nucleotide-induced conformations of the Switch loops (Fig. 3), we compared the structural relatedness of the core Thor-Rab domain after the removal of the Switch loops (Supplementary Table 2). 49 of the 69 top hits were Rab structures, 19 were K-Ras structures and 1 was a H-Ras structure, as ranked by the program Dali[29]. This confirmed that Thor-Rab is closest in structure to Rab and Ras family proteins, and more distant to other classes of eukaryotic GTPases (Arf), and even more distant to the bacterial GTPases (MglA and EngB). Inspection of structure-based sequence alignments revealed that Thor-Rab shows a high level of sequence homology in regions that have been used to define the eukaryotic Rab family (RabF1-F5, Fig. 1d)[3,26]. Thor-Rab has an arginine residue (Arg37) in an equivalent position in the sequence alignment to MglA Arg53 in Switch I (red triangle, Fig. 1d and Supplementary Fig. 2g). This residue in MglA has been predicted to act as an intrinsic "Arg finger" in stabilizing the GTP γ-phosphate during hydrolysis[8]. Other Rab paralogs from the Thor SMTZ1-45 genome do not have an arginine residue in this position (Supplementary Fig. 3). Thus, the MglA mechanism of γ-phosphate self-stabilization by arginine is not a common feature in Thor Rabs, implying that they may require GAPs to enhance hydrolysis. A phylogenetic tree calculated from a structure-based sequence alignment based solely on experimentally determined structures, placed SMTZ1-45 Thor-Rab closest to the Rab and Ras clades (Fig. 1e). Taken together with the high structural homology, this indicates that Thor Rabs are likely to be functionally related to Rab-like Ras GTPases. However, one of the defining characteristics of eukaryotic Rabs is the presence of one or two C-terminal cysteine residues which can be geranylgeranylated (Fig. 1d). These cysteine residues are absent from Thor-Rab and the C-terminus is significantly truncated relative to eukaryotic Rabs. Similarly, we could not find homologs for the eukaryotic Rab Escort Protein 1 and Rab geranylgeranyltransferase subunits in BLAST sequence databases searches. This indicates that Thor-Rab is not modified for insertion into membranes in the same manner as eukaryotic Rabs. As far as we know, Asgard archaea only have one membrane, the cell membrane[27,28]. Thus, the Rab cysteine-containing C-terminal extension likely arose in proto-eukaryotes in conjunction with the acquisition and distinction in internal membranes.

**Fig. 1 | Asgard archaea small GTPases. a** The X-ray structure of SMTZ1-45 Thor-Rab bound to GDP (sticks). The Rab core is shown in cyan with the switch I (red), switch II (orange), P-loop (blue), N-terminus (N), and C-terminus (C) regions highlighted. **b**, **c** Structural superimposition of SMTZ1-45 Thor-Rab (colored as in **a**) with a eukaryotic (**b**, human Rab-1B, green, PDB 4i1o[45]) and a bacterial (**c**, *T. thermophilus* MglA, brown, PDB 3t1o[8]) small GTPase. Switch I and Switch II in the overlaid structures are shown in pink and yellow, respectively. **d** Sequence alignment of the structures from **b**, **c** and the yeast Rab, Ypt1. The P-Loop, Switch I and Switch II are highlighted and colored as in Fig. 1a. The Rab-specific sequences are highlighted in pink (RabF1-RabF5)[26]. Triangles indicate two residues implicated in the catalytic mechanism for MglA[8]. The two C-terminal cysteine residues which can be geranylgeranylated in eukaryotic Rabs are highlighted in green. Stars indicate highly conserved residues in the Rab/GEF interface[3]. **e** A phylogenetic tree calculated from a structure-based sequence alignment based solely on experimentally determined structures. **f** Domain architectures, predicted by AF2, of the small GTPases sequences found in the MKD1 genome. G GTPase, R roadblock domain, L longin domain, "?" various domains of unknown function. **g**, **h** Phosphate release assay monitoring the GTPase activity of Rab11B and Thor-Rab, respectively, relative to the corresponding denatured GTPases.

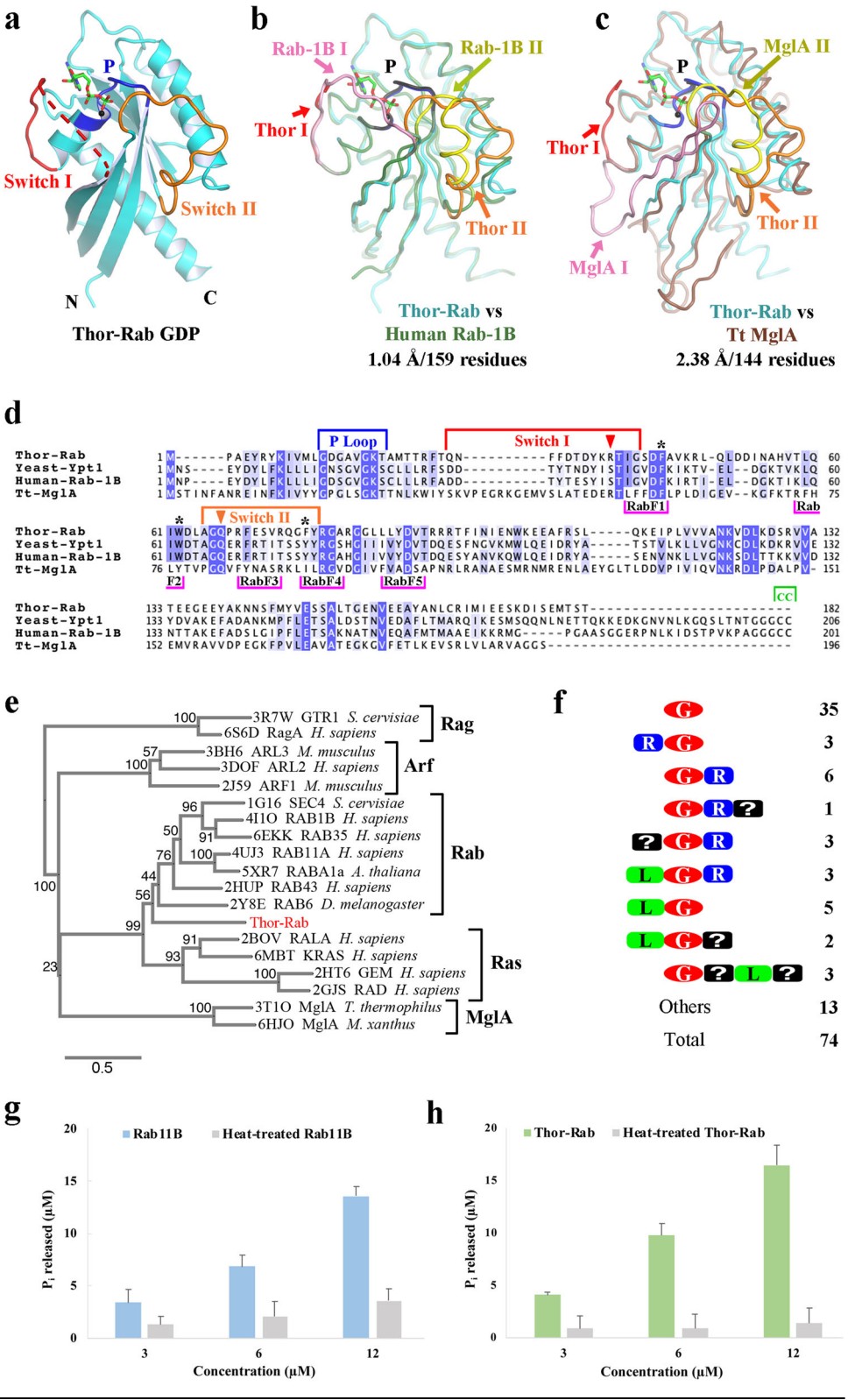

To measure the GTPase activity of SMTZ1-45 Thor-Rab, we employed a phosphate release assay[30]. SMTZ1-45 Thor-Rab generated similar levels of phosphate in comparison to human Rab11B (Fig. 1g, h) over two hours, whilst the heat-denatured proteins produced substantially less phosphate under the same conditions. This indicates that the unassisted GTPase activity of SMTZ1-45 Thor-Rab is similar to eukaryotic Rab proteins, implying that a GAP and/or GEF may

be needed to accelerate GTP hydrolysis and nucleotide exchange for signaling.

## Asgard small GTPases

To get a broader picture of the diversity of small GTPases in a single Asgard species, we searched the MKD1 genome, the first Asgard genome to be fully sequenced, and found 74 related sequences, for which we predicted the

**Fig. 2 | Comparison of the crystal packing around Switch I and II between the two molecules of Thor-Rab in the asymmetric unit. a, b** Contacts around Switch I. **c, d** Contacts around Switch II. In each case, the center molecule is colored as in Fig. 1a. Neighboring molecules are shown in brown. Crystal contacts with Switch I and Switch II are indicated by asterisks, colored red and orange, respectively. The two molecules in the asymmetric unit display different crystal contacts for Switch I and Switch II, which influences the conformations of these regions and the nucleotide accessibility.

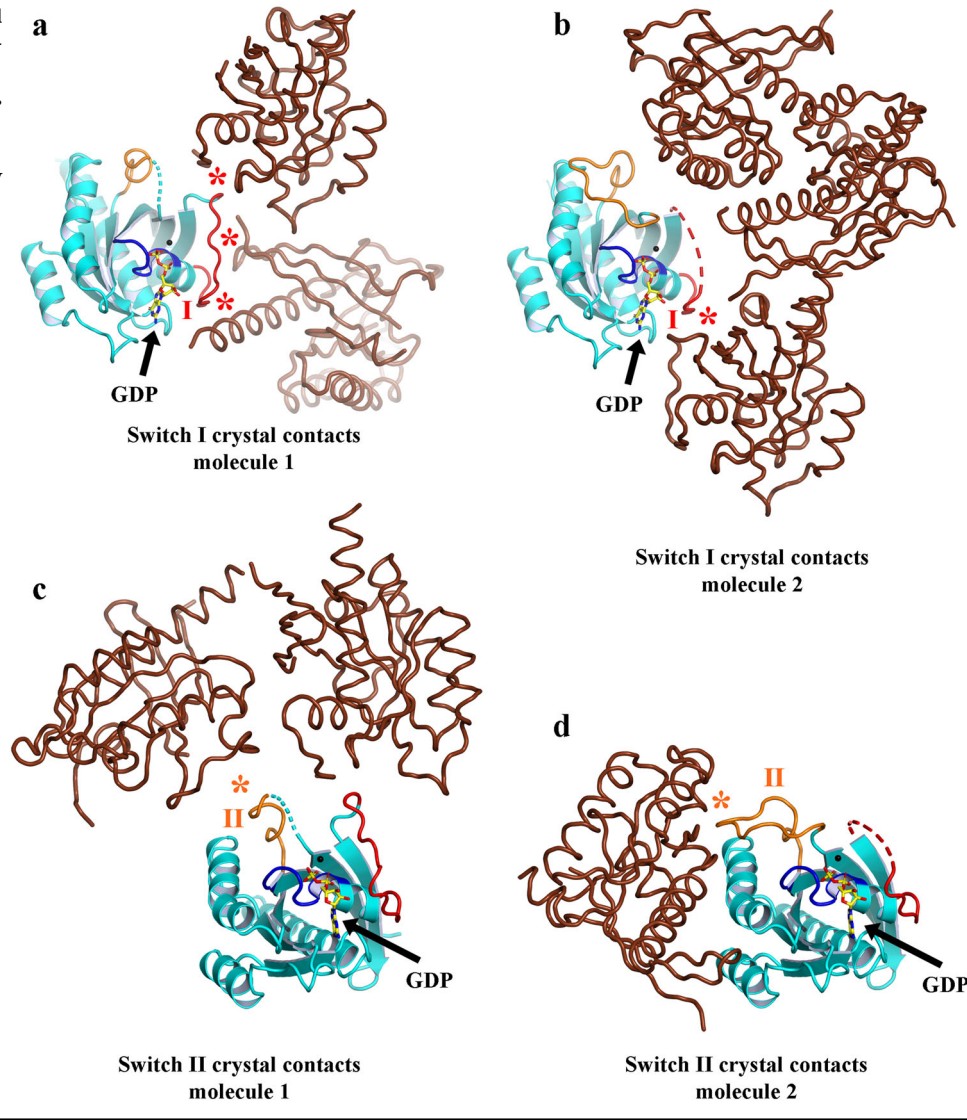

structures using AlphaFold2 (AF2, Fig. 1f). 35 of the sequences consisted of single domain GTPases and 26 sequences are combined with RB or longin domains (Fig. 1f). We calculated a phylogenetic tree from a structure-based sequence alignment of the small GTPase domains and included a variety of eukaryotic and prokaryotic structures (Supplementary Fig. 4 and Supplementary Table 3). Onto this tree, we mapped the domain architectures and the GTPase classes calculated from the superposition of AF2 models of the GTPase domains onto experimentally determined structures (Supplementary Fig. 4). This revealed the major class of GTPases to be Rab-like GTPases, as predicted from sequence annotation (Supplementary Figs. 4 and 5). The second largest class was most similar to Rag-like GTPases, but also close to Arf GTPases in structural homology (Supplementary Figs. 4 and 6). Many of these proteins had been predicted to be Arf GTPases from sequence annotation. These architectures include the GTPase domain fused to a longin or to a RB domain, or fused between a longin domain and RB domain. Interestingly, an insert was observed in Switch II from GTPase-RB architectures but not in RB-GTPase architectures, indicating potential differences in nucleotide regulation between these designs (Supplementary Fig. 6a). Finally, a smaller group contained AF2-predicted MglA bacterial-like GTPases (Supplementary Figs. 4 and 7). One pair of MKD1 Rag-like GTPase genes exist in an operon in the genome. The AF2-predicted structure reveals a potential heterodimer to be connected via the RB domains in a similar way to eukaryotic Rag-like complexes, such as the yeast GTR1/GTR2 heterodimer (Supplementary Fig. 8a–d)[31]. AF2-predicted

structures of homodimers of other Rag-like GTPase architectures (Supplementary Fig. 8e–h) indicate a variety in the position of the GTPase domain relative to the longin/RB domains. Thus, from AF2 modeling, the MKD1 genome is predicted to encode multiple paralogs of Rab, Rag, and MglA GTPases, proteins which have membrane-associated roles in eukaryotes and in bacteria.

## RB

Subsequently, we investigated a predicted RB protein (KXH72322.1) from Thor SMTZ1-45, and a second RB protein (WP_147663254.1) from Loki MKD1 for comparison with the Loki profilin structures[32]. These proteins were chosen based on sequence homology to eukaryotic RB domains. The proteins (Thor-RB and MKD1-RB) were purified, crystallized and their structures were elucidated via X-ray crystallography to 2.14 and 2.69 Å, respectively (Fig. 4a, b and Supplementary Table 4). Both proteins are homodimers formed from protomers comprised of 5-stranded β-sheets sandwiched between a single α-helix (formed between strands 2 and 3) and a pair of α-helices (formed from the N- and C-termini) (Fig. 4a–c, f).

We compared the structural similarities of Thor-RB and MKD1-RB with MglBs and eukaryotic RB proteins. Both proteins displayed approximately similar levels of homology with the RB proteins *M. xanthus* MglB and *H. Sapiens* LAMTOR2 (Fig. 4g–j and Supplementary Table 5), matching ~110 residues with an RMSD of 1.5–1.6 Å. Lower levels of homology were seen in comparison with longin domains (86–94 residues,

**Fig. 3 | Comparison of Thor-Rab with eukaryotic Rab structures containing similar nucleotides.** **a, b** GDP-bound Thor-Rab (7EZB) compared to *Arabidopsis thaliana* RabA1a (GDP; 5xr4), human Rab1A (GDP; 2fol), human Rab8 (GDP; 4lhv). **c–f** GTPγS-soaked Thor-Rab crystal structures (**c, d** 7EZD and **e, f** 7EZE) compared to human Rab4A (GNP; 1yu9), human Rab8 (GNP; 4lhw), and mouse (GNP; 1z06). The Rab core structures are very similar. Variation is seen in the Switch I and II, which is influenced by crystal packing for Thor-Rab.

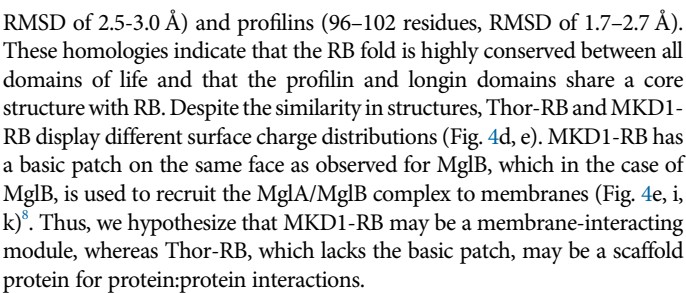

RMSD of 2.5-3.0 Å) and profilins (96–102 residues, RMSD of 1.7–2.7 Å). These homologies indicate that the RB fold is highly conserved between all domains of life and that the profilin and longin domains share a core structure with RB. Despite the similarity in structures, Thor-RB and MKD1-RB display different surface charge distributions (Fig. 4d, e). MKD1-RB has a basic patch on the same face as observed for MglB, which in the case of MglB, is used to recruit the MglA/MglB complex to membranes (Fig. 4e, i, k)[8]. Thus, we hypothesize that MKD1-RB may be a membrane-interacting module, whereas Thor-RB, which lacks the basic patch, may be a scaffold protein for protein:protein interactions.

## $RB_{LC7}$

Next, we explored truncated RB sequences within the Asgard sequence databases. One potential $RB_{LC7}$ sequence from Odin LCB_4 (OLS18093.1,

Odin-$RB_{LC7}$) was amenable to protein expression, purification, and structure determination by X-ray crystallography, refined against 1.83 Å data (Fig. 5a, b and Supplementary Table 4). The Odin-$RB_{LC7}$ structure shares a similar homodimeric structure to Thor-RB and MKD1-RB but lacks the terminal pair of α-helices observed in the RB structures (Fig. 4a, b). However, the interpretable electron density starts at Gln13 and the preceding 12 residues are predicted to form an α-helix by AF2 (Fig. 6a, b and Supplementary Fig. 9a). Thus, Odin-$RB_{LC7}$ forms a $RB_{LC7}$ conformation which lacks the C-terminal α-helix found in the longer RB architectures.

We compared the Odin-$RB_{LC7}$ structure to other RB, $RB_{LC7}$, longin and profilin structures (Supplementary Table 5). Odin-$RB_{LC7}$ was most similar to MglB and dynein light chain RB domain 1 (DLRB1) homodimers (77–78 residues, RMSD of 1.3–1.4 Å, Fig. 5c) and showed good homology to the LAMTOR4/5 heterodimer (71 residues, RMSD of 1.7–1.8 Å, Fig. 5e) and

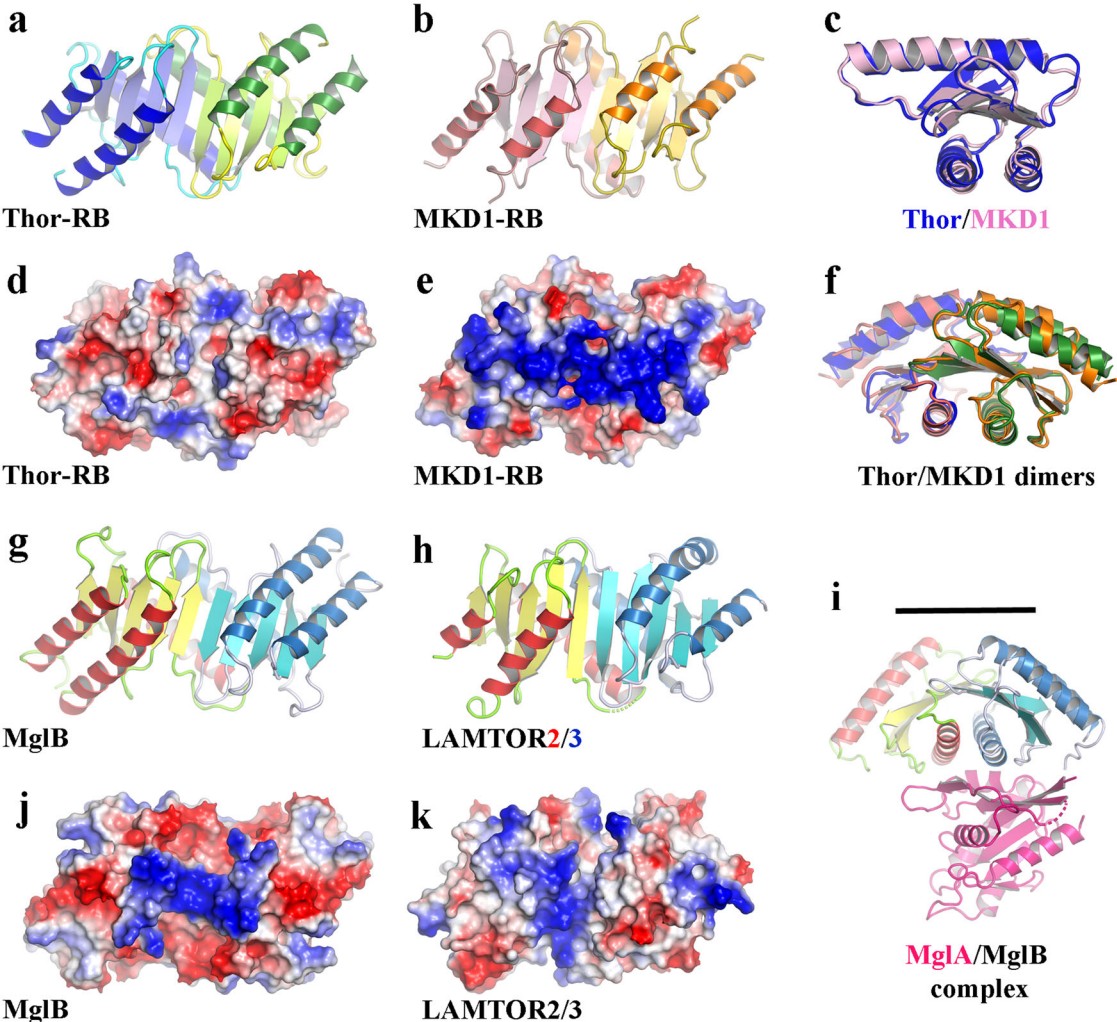

**Fig. 4 | The X-ray structures of RB proteins. a** Thor-RB homodimer. **b** MKD1-RB homodimer. **c** Overlay of Thor-RB and MKD1-RB protomers. **d** Thor-RB surface charge. **e** MKD1-RB surface charge. **f** Overlay of Thor-RB and MKD1-RB dimers, colored as in **a**, **b**. **g** MglB homodimer (PDB ID 3t1s)[8]. **h** LAMTOR2/3 heterodimer (PDB ID 5y3a)[46]. **i** MglB surface charge. **j** LAMTOR2/3 surface charge. **k** MglA/MglB complex (PDB ID 3t1q)[8]. MglA is colored in pink, MglB as in **g**. The line indicates the membrane interaction surface.

Odin profilin[32] (64 residues, RMSD of 1.8 Å). Comparison of the $RB_{LC7}$ structures in their larger complexes reveals that the absent C-terminal α-helix, relative to RB structures, allows for the association of an α-helix from a binding partner, DC1I2 for DLRB1 (Fig. 5d) and LAMTOR1 for LAMTOR2/3 (Fig. 5f). We propose that the Odin-$RB_{LC7}$ homodimer will act in a similar manner in providing an α-helix binding site to assemble larger complexes.

## Longin

We were not successful in solving the structure of a longin domain, however many such sequences are predicted in the Asgard genomes[22,23]. We used AF2 to explore the structure of a longin domain protein from MKD1 (Supplementary Fig. 9b), which was predicted to have a similar fold to eukaryotic longin domains.

## Comparison of RB/longin/profilin folds

Comparison of the structures and topologies of the RB, longin and profilin families of proteins indicates the adaptations in these proteins that surround the core domain (Figs. 6 and 7). $RB_{LC7}$ has lost the C-terminal helix relative to RB, while the longin domain has lost the N-terminal helix and instead placed an additional C-terminal helix at the same location (Fig. 7a–c). Whereas, half of the central helix is replaced by a 3-strand motif in profilin (Fig. 7d). This region is responsible for dimerization in RB proteins (Fig. 7e, f) and in actin

binding of profilin (Fig. 7g, h). Thus, the core common fold is comprised of a 5-stranded β-sheet surrounded by adaptable α-helices that mediate interactions with different binding partners. It is likely that differentiation in these proteins occurred in the ancestors of Asgard archaea, since these proteins are found in all Asgard archaea phyla[33], and expansion in numbers of each fold continued in the subsequent Asgard lineages leading to the variations between lineages, as documented for profilins[34,35].

## TRAPPC3

Finally, we investigated the TRAPPC3-like proteins from Thor. We expressed, purified, and crystallized two potential TRAPPC3 proteins (Thor-TRAPPC3), from Thor SMTZ1-45 (KXH75250.1) and Thor AB25 (OLS30461.1). The crystals were pink in color indicating potential $Zn^{2+}$ binding. We scanned $Zn^{2+}$ fluorescence for AB25 Thor-TRAPPC3 using X-rays to confirm the identity of the cation and collected multiple anomalous dispersion (MAD) diffraction data around the $Zn^{2+}$ edge to solve the structure at 1.91 Å (Supplementary Table 6 and Supplementary Fig. 10a, b). Subsequently, SMTZ1-45 Thor-TRAPPC3 was solved at 1.7 Å resolution, by molecular replacement using the AB25 Thor-TRAPPC3 structure (Supplementary Table 6). We concentrated on the analysis of the SMTZ1-45 Thor-TRAPPC3 structure since it was refined against higher resolution data.

SMTZ1-45 Thor-TRAPPC3 forms a homodimer that closely resembles the eukaryotic heterodimer of TRAPPC3/C6 (Fig. 8a–c). The SMTZ1-45

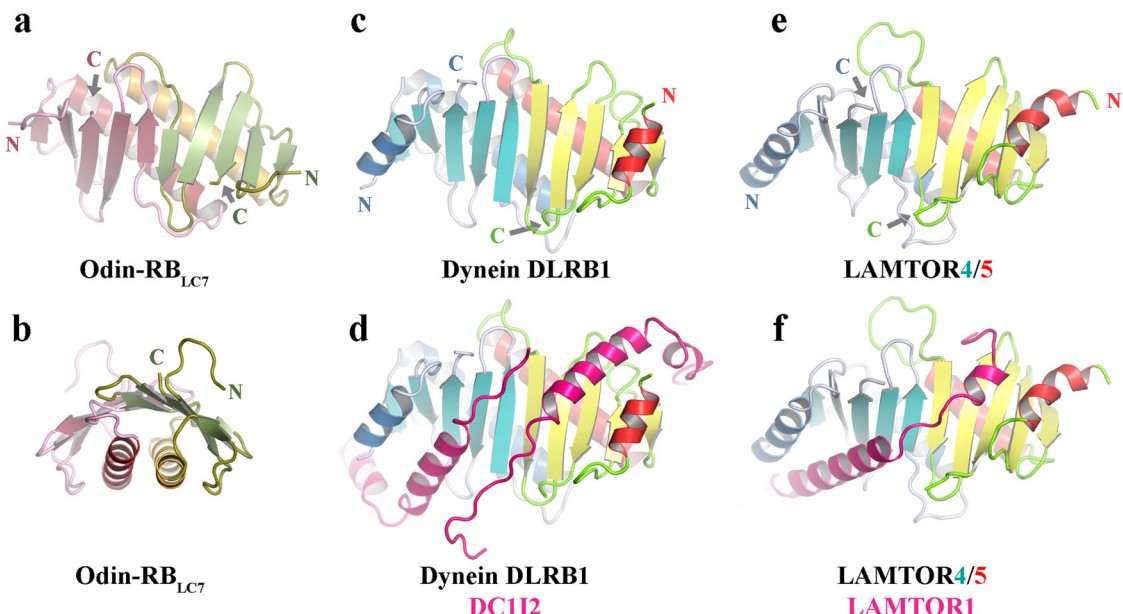

**Fig. 5 | The X-ray structures of RB$_{LC7}$ proteins. a, b** Two views of the Odin-RB$_{LC7}$ homodimer. **c** Dynein DLRB1 homodimer alone and (**d**) in complex with DC1I2 (PDB ID 6f1z6)[47]. **e** LAMTOR4/5 alone, and (**f**) in complex with LAMTOR1 (PDB ID 5y3a)[46]. The N- and C-termini are indicated for the protomers in **a, c, e**.

Thor-TRAPPC3 subunit is most closely structurally related to TRAPPC3, characterized by 148-151 matching residues with RMSDs of 2.1–2.3 Å (Supplementary Table 7). The SMTZ1-45 Thor-TRAPPC3 architecture forms two layers, comprised of 4 α-helices and 4-stranded β-sheet, with the two N-terminal helices forming the dimerization interface. Bacterial V4R domains, such as the PoxR homodimer, are also structurally similar to SMTZ1-45 Thor-TRAPPC3 (Fig. 8b, d and Supplementary Table 7, 130 matching residues, RMSD of 3.3 Å), albeit more distant than eukaryotic TRAPPC3/C6. SMTZ1-45 Thor-TRAPPC3 also shows structural homology to other ligand-binding proteins, such as the bacterial NO-binding heme-dependent sensor protein (H-NOX) and cellulose synthase subunit D (AxCeSD), and to human soluble guanylate cyclase (GUCY1A/B) (Supplementary Table 7). Besides sharing a common fold and dimerization geometry, SMTZ1-45 Thor-TRAPP also contains internal cavities (Fig. 8b). Such cavities bind to hydrophobic ligands for mouse TRAPPC3 (palmitic acid, Fig. 8c)[36] and PoxR (phenols, Fig. 8d)[37], indicating that the function of binding ligands is maintained during evolution. We speculate that Thor-TRAPPC3 may also bind small molecules, potentially being involved in ligand transport. The cavities appear to be accessible from the exterior close to the dimerization interfaces (Fig. 8e–g). SMTZ1-45 Thor-TRAPPC3 Zn$^{2+}$ binding occurs through 4 cysteine residues (Fig. 8b), similarly to PoxR (Supplementary Fig. 10c). We used this feature to search for TRAPP domains in other Asgard phyla. We found sequences in MKD1 and Heimdall that were predicted by AF2 to adopt the TRAPPC3/V4R fold and to form homodimers (Fig. 9 and Supplementary Fig. 10d–f)[38]. Interestingly, we identified two MKD1 proteins, one of which was predicted by AF2 to be more similar to Thor-TRAPPC3 (Fig. 9a, b) and the other more similar to the PoxR V4R domain (Fig. 9e, f) in their topologies and potential Zn$^{2+}$ binding. We further analyzed the homology around the Thor-TRAPPC3 Zn$^{2+}$-binding site (Fig. 9). We found that the predicted Zn$^{2+}$-binding sites (Fig. 9b,d, e) and the experimentally determined sites (Fig. 9a, f) are not completely conserved in positions of the coordinating residues, rather they appear in similar regions and appear to serve the same function in tethering the β-sheet to the D α-helix to create cavities. Interestingly, mouse TRAPPC3 has evolved to replace the Zn$^{2+}$-binding site by a hydrogen bond (Figs. 8c and 9c). Taken together, our structural and sequence analyses of Asgard TRAPP/V4R proteins suggest that the potential role of these proteins will be in ligand binding.

Finally, we expressed the Asgard proteins from this study as GFP fusion proteins in human HeLa cells. The GFP signal under a fluorescent microscope was observed to be diffuse throughout the cytoplasm and nucleus for all Asgard proteins tested (Supplementary Fig. 11). We did not observe targeting of the Asgard proteins to membranes.

## Discussion

In summary, our structural and sequence analyses indicate that Thor-archaeota and MKD1 contain primordial Rab proteins that have the core structure of eukaryotic Rabs but lack the C-terminal extension, which in eukaryotes is geranylgeranylated for membrane insertion. Since internal membranes were not observed in the two Asgard archaea isolated to date[27,28], we speculate that the C-terminal extension arose during eukaryogenesis, and, perhaps not surprisingly, we saw no membrane localization of Thor-Rab when expressed in eukaryotic cells. We found no small GTPase sequence in MKD1 that contained terminal cysteine residues, indicating that these proteins are not modified in the same manner as eukaryotic Rabs for membrane insertion, and suggesting that the eukaryotic small GTPases will have undergone gene duplication events after the evolution of these features. The role of the Rab and Rag proteins in Asgard archaea is currently unknown. We hypothesize that the large number of these sequences found in the Asgard databases indicate that these organisms are able to specify different areas of the membrane, possibly using the Rab-like and Rag-like GTPases for regulation of protrusions or smaller compartments such as vesicles, which have been observed in Loki[27,28].

We also found structural evidence that Asgard archaea contain RB and RB$_{LC7}$ proteins. These two designs are combined in the eukaryotic Ragulator-Rag complex (Fig. 10). Furthermore, we uncover structural evidence that TRAPPC3-like proteins exist in several branches of Asgard archaea and are likely to share a common ancestor gene with the V4R-like domain. We modeled a TRAPPC1-like longin domain from MKD1 (Supplementary Fig. 9b). Thus, Asgard archaea have all the core protein folds that are found in eukaryotic TRAPP complexes (Fig. 10). One longin-domain sequence from Heimdall combines a TRAPPC3 domain with the longin domain (Supplementary Fig. 10g). Other domain combinations, in MKD1, fuse RB or longin domains with GTPases (Fig. 1f). These architectures indicate that functional interactions between RB and GTPases and between longin, TRAPPC3 and GTPases exist in Asgard archaea. However, MK-D1 small GTPase genes do not generally exist within operons with RB, longin, or TRAPPC3 genes. This indicates that the use of operons is not a major method of regulation of GTPase/regulator co-expression, and

**Fig. 6 | Two views of the structure of the Thor-RB protomer (blue) superimposed onto related structures. a, b** Superimposed onto the Odin-RB$_{LC7}$ structure (pink) and AF2 prediction (yellow). The pink asterisk highlights Gln13, the first ordered residue in the X-ray structure. The C-helix observed in Thor-RB is missing from Odin-RB$_{LC7}$ (blue arrow). **c, d** onto the AF2 prediction of MKD1 longin protomer (green), taken from the homo-dimer prediction. **e, f** onto the Loki profilin structure (brown).

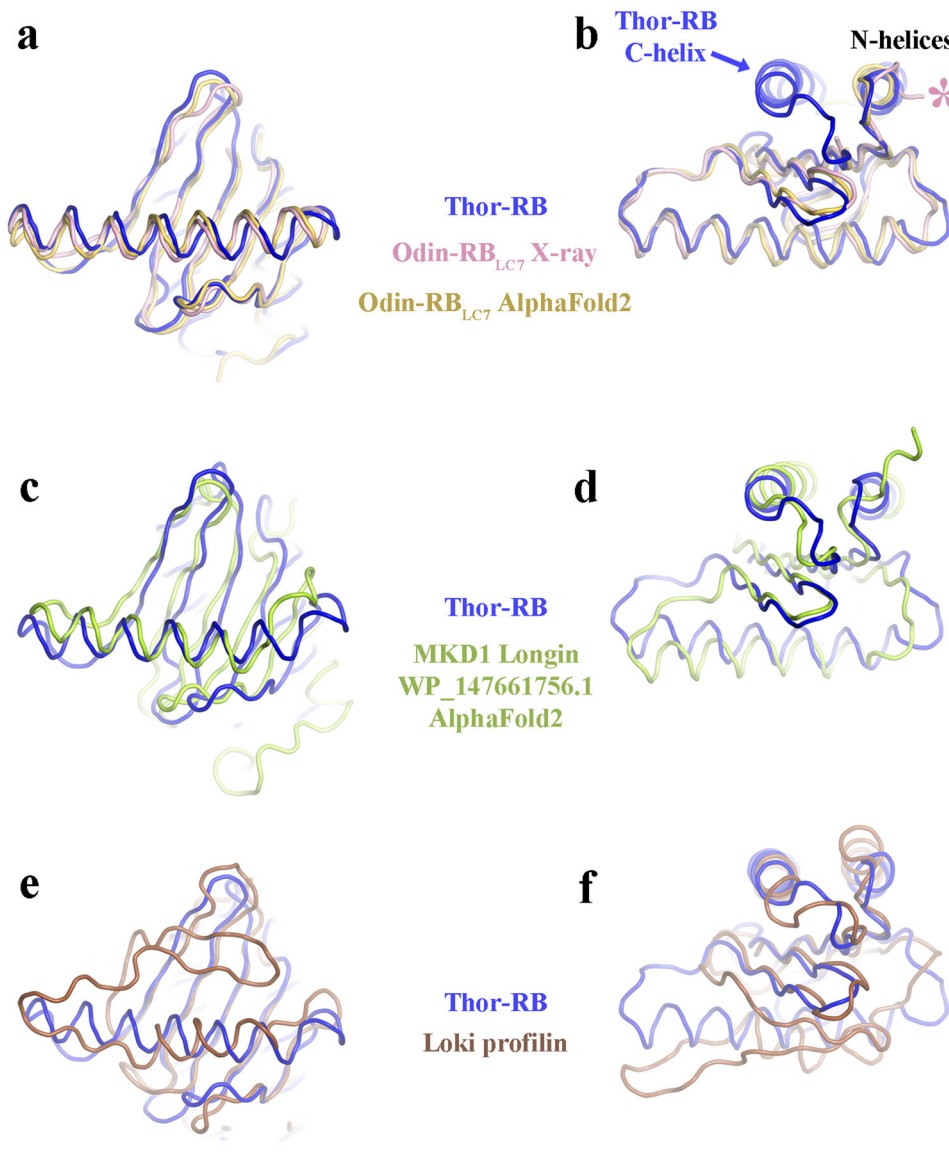

perhaps suggests that transcription factor specific regulation may be used. This makes matching of individual small GTPases to potential GAPs or GEFs difficult to predict from the genome lay out. We conclude that Asgard archaea possess multiple copies of eukaryotic-like components of GTPase membrane signaling complexes (Fig. 10), indicating that these archaea likely have highly sophisticated membrane regulation.

## Methods
### Protein expression and purification
Asgard Thor-Rab, RB and TRAPPC3 gene sequences and the human Rab11B gene sequence were codon-optimized (*Escherichia coli*), synthesized (GenScript), and subcloned into the pSY5 vector including an N-terminal HRV 3 C protease cleavage site and 8-histidine tag. Thor-Rab was initially synthesized in a pEX-A2J2 vector (Eurofins) and subcloned in the pSY5 vector. Asgard proteins and human Rab11B were expressed as described[25]. The cell pellets were extracted with binding buffer (20 mM HEPES, pH 7.5, 500 mM NaCl, 20 mM imidazole) supplemented with 0.01% TritonX-100 (Nacalai), protease inhibitor cocktail (EDTA-free, Calbiochem) and 2 µl of 10,000 u/µl benzonase (Merck). Cell lysis was performed using an ultrasonic cell disruptor (Branson) with 5 s of pulse, 30-40% duty for 5 min. Proteins were loaded on a Ni-NTA affinity chromatography column (Qiagen), and washed with five column volumes of binding buffer. The N-terminal His-tag was removed by

cleavage with HRV-3C protease at 4 °C, overnight. The eluted proteins in the binding buffer were subjected to a size exclusion chromatography (Enrich SEC 70, Bio-Rad) in 20 mM HEPES, pH 7.5, 150 mM NaCl. Proteins were pooled and concentrated with 10 kDa MWCO centrifuge filters (Merck).

### Crystallization
Asgard proteins at ~ 10 mg/ml were exchanged and stored in 10 mM HEPES, pH 7.5, 30 mM NaCl. Protein crystallization trials were carried out on a Gryphon LCP robot (Art Robbins Instruments) using the sitting-drop or vapor-diffusion methods with PACT-primer and JCGS-plus commercial screens (Molecular Dimensions) (1:1 protein:reservoir) in a 96-well Vio-lamo crystallization plate (As one) at 20 °C. Thor-Rab crystals were obtained in 100 mM MES, pH 5.5, 250 mM NaCl, 13% wt/vol PEG 8000. Crystals for structure 7EZB included 5 mM GTP and 5 mM GDP in the crystallization mix. Crystals for structure 7EZD included 10 mM GTPγS in the crystal-lization mix. Crystals for 7EZE were formed under the same conditions as 7EZB but were soaked (30 min) with 10 mM GTPγS and 15 mM MgCl$_2$ prior to harvesting. Thor-RB formed crystals in 100 mM sodium acetate trihydrate, pH 5.0, 200 mM LiCl and 16% wt/vol PEG 6000. MKD1-RB was crystallized in 100 mM MES, pH 6.0, 250 mM zinc acetate dihydrate, and 6% wt/vol PEG 3000. OdinRB$_{LC7}$ crystals were grown in 200 mM MgSO$_4$, 20% PEG 4000 and 10% glycerol. Thor-TRAPPC3 crystals were generated

**Fig. 7 | Adaptation of the RB fold in Asgard archaea and eukaryotes. a–d** Topology cartoons of typical architectures. Common β-strands and α-helices are shown in yellow and orange. Differences with respect to profilin are colored blue. **a** RB has N- and C-terminal helices. **b** RB$_{LC7}$ is missing the C-terminal helix. **c** Longin is missing the N-terminal helix, but replaces it with a second C-terminal helix (pink). **d** Profilin has a 3-strand motif replacing the C-terminal half of the central helix (blue). **e** RB monomer and (**f**) RB dimer. The C-terminal half of the central helix (blue) mediates dimerization in RB proteins (blue arrow). In the dimer structure (**f**) of one protomer is colored as in A, the other protomer is colored pink. **g** Profilin/actin complex and (**h**) profilin alone. The profilin 3-strand motif (blue) is involved in actin (cyan) interaction (PDB ID 5yee) indicated by the blue arrow[32]. The structure of profilin is colored as in **d**.

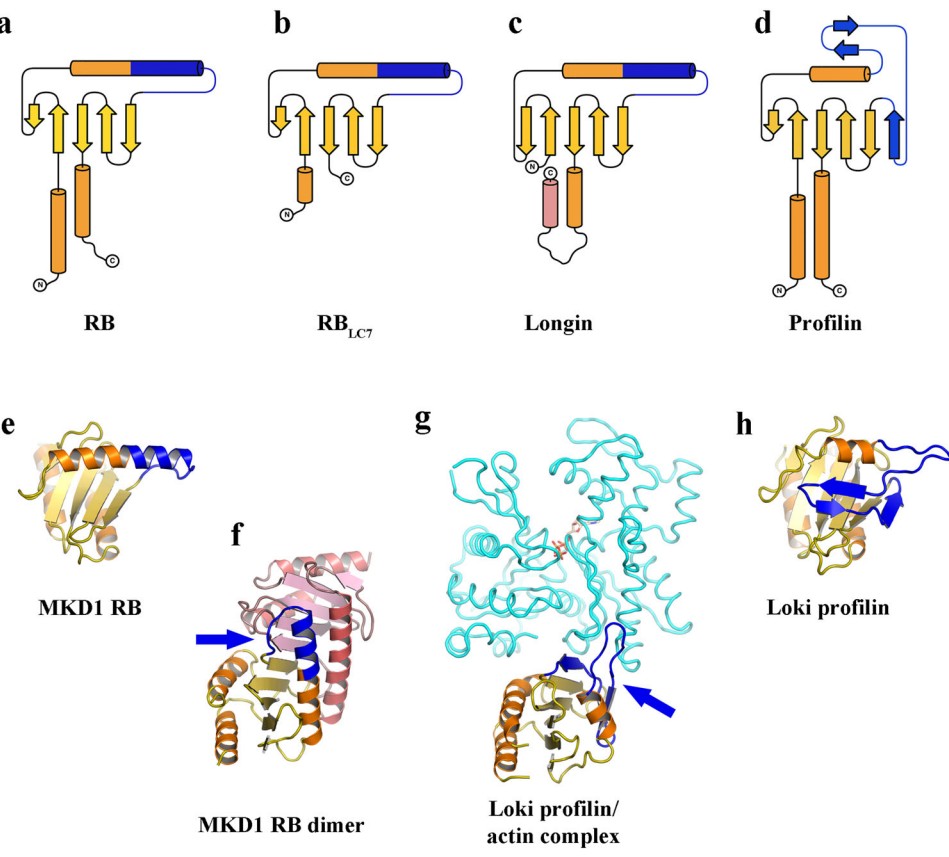

**Fig. 8 | Comparison of SMTZ1-45 Thor-TRAPPC3 with eukaryotic TRAPPC3 and bacterial V4R structures. a** A dendrogram of structural similarities indicating that SMTZ1-45 Thor-TRAPPC3 is most similar to eukaryotic TRAPPC3. **b** The structure of the SMTZ1-45 Thor-TRAPPC3 homodimer (pink, light blue) with cavities (yellow). Expanded regions indicate the cavity (charge surface) and Zn$^{2+}$-binding site (black sphere). **c** The eukaryotic TRAPPC3/C6 complex bound to palmitic acid (PDB ID 2j3t)[36]. **d** The core of the V4R domain from bacterial PoxR bound to a phenol derivative (PDB ID 5fs0)[37]. **e–g** A slice through structures from **b–d** surrounded by the charged surface. Arrows indicate the potential entry to access the cavities.

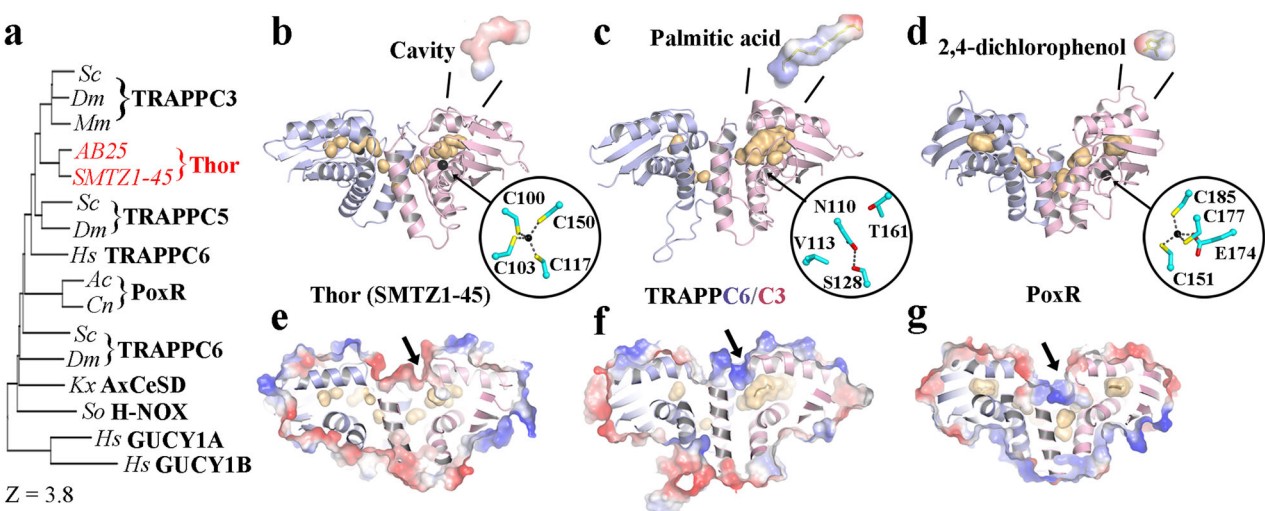

in 100 mM HEPES, pH 7.5, 250 mM magnesium formate-dihydrate, and 18% wt/vol PEG3350. Crystals were flash frozen in the crystallization buffer prior to data collection.

**Structure determination, model building, and refinement**
X-ray data were collected on RAYONIX MX-300 HS CCD detector on beamline TPS 05 A (NSRRC, Taiwan, ROC) at λ = 1.0 Å or on BL41XU (λ = 1.0 Å) SPring-8 on a Pilatus 6 M detector. Data were indexed, scaled, and merged following standard protocols[32]. Molecular replacement and refinement were carried out using PDB 3BFK as the search model using standard methods to solve the structure 7EZB. Subsequently, the structures of 7EZD and 7EZE were solved using 7EZB as the starting model. The identities of the bound nucleotides were assessed by refinement of GTP, GDP or a combination of GTP and GDP in the nucleotide-binding

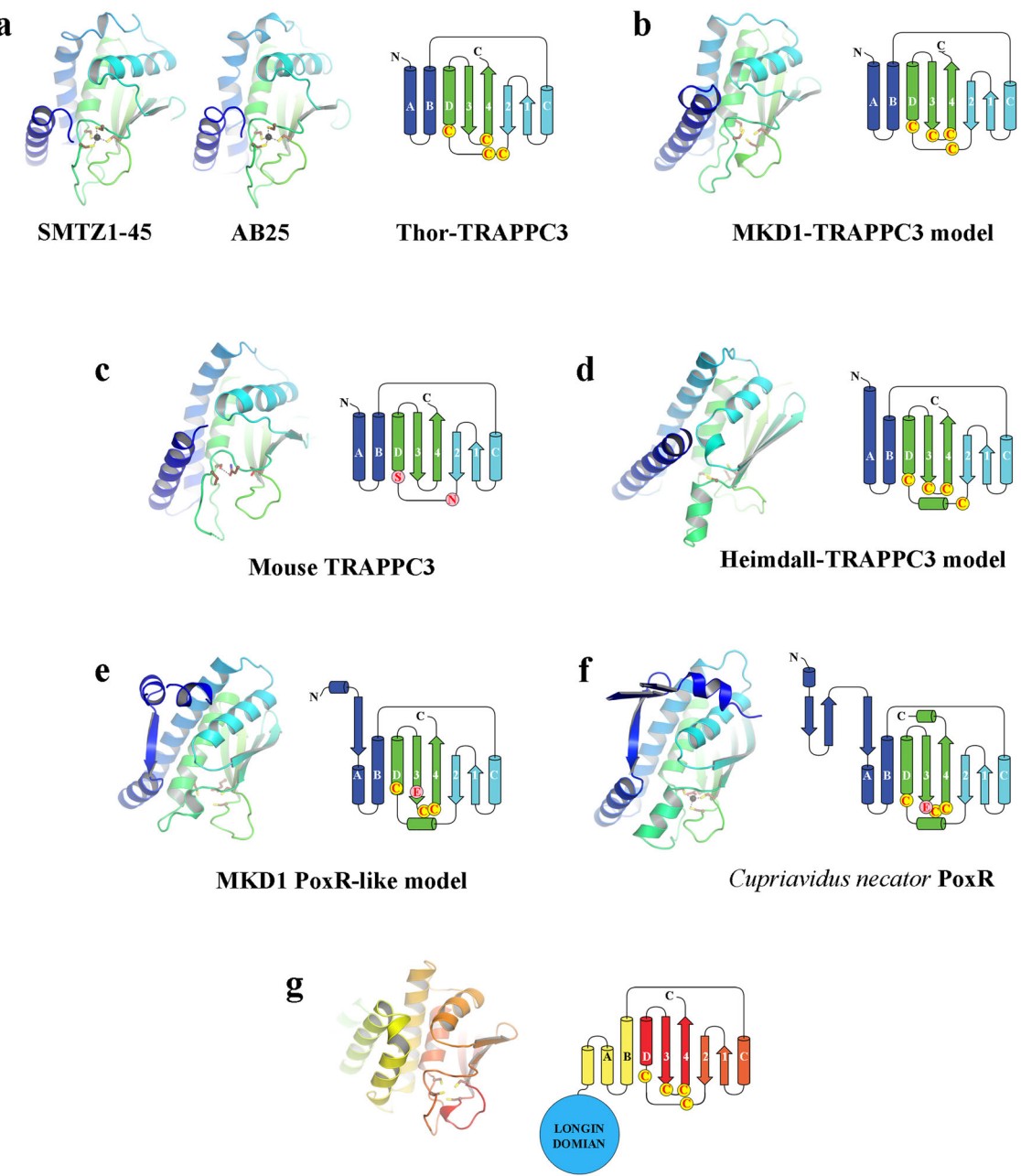

**Fig. 9 | Comparison of Thor-TRAPPC3 with other structures and AF2 models.**
**a–e** Protein structure (**a**) or AF2 models (**b–e**) are shown next to topology diagrams. The color scheme is blue to green, N- to C-termini. Colored circles indicate residues in Zn²⁺-binding sites, determined or predicted. **b** MKD1-TRAPPC3 (WP_147663132.1). **c** The mouse TRAPPC3 structure has a hydrogen bond between Ser and Asn in the same region. **d** Heimdall-TRAPPC3 (PWI47524.1). **e** MKD1 PoxR (WP_147661794.1). **f** PoxR structure (PDB ID 5FS0). **g** An AF2-predicted model of the Heimdall protein (MBD3190748.1) that comprises an N-terminal longin domain (LG) and a C-terminal TRAPPC3 domain, from Supplementary Fig. 10g.

sites. Odin-RB$_{LC7}$ and Thor-RB were solved by molecular replacement using the dynein DLRB1 structure (PDB 3L7H) and MglB (PDB 3T1S), respectively. MKD1 was solved using the Thor-RB structure. Standard refinement and building protocols were implemented as described for OdinTubulin[39]. The Thor-TRAPPC3 (AB25) structure was elucidated via a three wavelength MAD experiment using the natively bound Zn²⁺. The initial model was constructed in AutoSol in PHENIX[40]. The refined model (7YH2) was used as the molecular replacement model for Thor-TRAPPC3 (SMTZ1-45). All models were refined in PHENIX and rebuilt in COOT[40,41].

## Sequence and structure analysis
Sequence alignment and phylogenetic analysis were carried out in MAAFT[42] and structural comparisons in Dali[29] and PDBeFold[43]. AlphaFold2 models were constructed in either monomer or multimer modes[38,44].

## Phosphate release assay
The GTPase activity of human Rab11B and SMTZ1-45 Thor-Rab were measured by quantifying the amount of inorganic phosphate (P$_i$) released by the enzymatic reaction using the EnzChek Phosphate Assay Kit (Molecular Probes)[30]. Different protein concentrations of Rab11B and

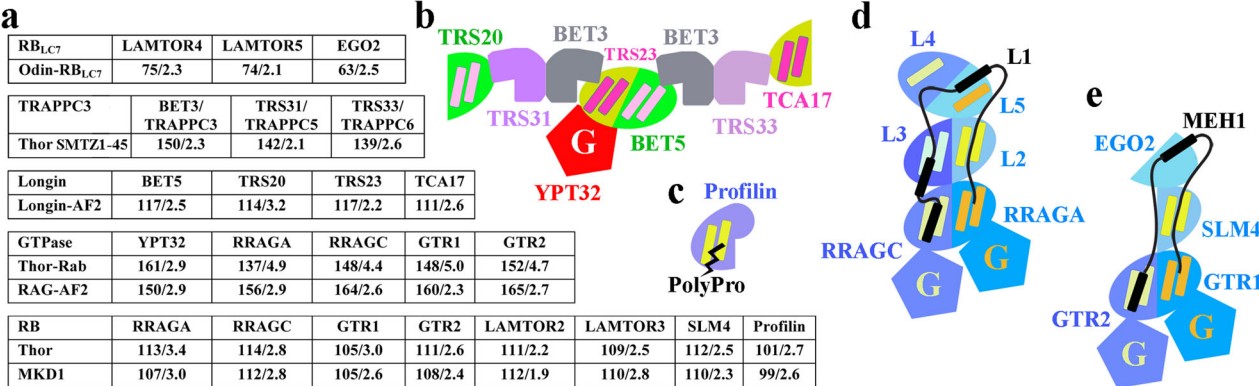

**Fig. 10 | Structural similarities of Asgard archaea and eukaryotic complex components. a** Structural comparison of Asgard archaea proteins with proteins from the TRAPP II, Ragulator-Rag, EGO, and profilin complexes. Similarities are indicated by the number of matching amino acids/RMSD in Ångströms. **b–e** Cartoon representations of the TRAPP II (PDB ID 7e8t)[48], profilin (PDB ID 2pdb)[49], Ragulator-Rag (PDB ID 6wj2)[50], and EGO (6jwp)[51] complexes, respectively. The pairs of helices on one face of the RB and profilin domains (one helix in the case of RB_LC7 domains) provide sites for protein:protein interactions (black in **c–e**). The pairs of helices on the longin domains of TRAPP II face the membrane.

SMTZ1-45 Thor-Rab were incubated at 25 °C for 2 h in 20 mM HEPES, pH 7.5, 150 mM NaCl, 1 mM GTP, and 2 mM MgCl$_2$. The reaction mixtures (50 µl) were then incubated with EnzChek Phosphate reagent (0.2 mM) at 25 °C for 30 min and the absorbance was measured at 360 nm (Infinite 200 PRO, Tecan). Heat-treated control samples were prepared at 95 °C for 15 min and centrifuged for 10 min, 20 °C at 20,000×$g$, and were assayed with the same protocol. A standard curve for the GTPase assay was created by adding from 2 to 20 µM of phosphate working solution to the standard reaction mixture EnzChek Phosphate reagent (0.2 mM). After incubating for 30 min at 25 °C, the P$_i$ standard absorbances were measured at 360 nm. Data analysis was generated by subtracting the value determined in the absence of Rab11B and SMTZ1-45 Thor-Rab proteins. GTPase activity experiments were performed in triplicate with the mean values and their standard deviations calculated.

## Cell culture and transfection
HeLa cells were cultured in Minimum Essential Media (MEM, Sigma-Aldrich) supplemented with L-glutamine and 10% fetal bovine serum (FBS) (Nichirei), and incubated at 37 °C with 5% CO$_2$. Mycoplasma contamination in cell cultures was routinely tested using the PCR mycoplasma detection set (Takara Bio). The EGFP-tagged constructs were transfected into cells using the Xfect transfection reagent (Takara Bio). After 24 h incubation, cells were fixed with 4% paraformaldehyde (Nacalai Tesque, Inc.) in PBS for 15 min at room temperature, mounted with Fluoro-KEEPER antifade reagent with DAPI (Nacalai Tesque, Inc.), and observed under an FV1200 confocal laser scanning microscope (Olympus).

## Statistics and reproducibility
The phosphate release assay was repeated 3 times with similar results.

## Data availability
The atomic coordinates and structure factors have been deposited in the Protein Data Bank under the accession codes: 7EVB-D, 7EVG-L, 7F1A-B, and 7YH1-3. All other data are available in the main text or the supplementary materials.

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

## Acknowledgements

We thank the Synchrotron Radiation Protein Crystallography Facility of the National Core Facility Program for Biotechnology, Ministry of Science and Technology and the National Synchrotron Radiation Research Center, a national user facility supported by the Ministry of Science and Technology, Taiwan, ROC, and the SPring-8 Synchrotron, Japan. This work was supported by JST CREST, grant number JPMJCR19S5 (R.C.R.); Japan Society for the Promotion of Science (JSPS), grant numbers JP20H00476 and JP22H04985 (R.C.R.), and JP19K23727, JP23K05718 and JP23H04423 (Y.S.); and by the Moore-Simons Project on the Origin of the Eukaryotic Cell, grant number GBMF9743 (R.C.R.); and Wesco Scientific Promotion Foundation (Y.S.).

## Author contributions

L.T.T., C.A., and R.C.R. conceptualized and designed research, and carried out structural biology research. L.T.T. carried out the biochemical assays. L.T.T. and Y.S. carried out cell biology research. R.C.R. and L.T.T. wrote the original draft of the manuscript. All authors read and approved the final manuscript.

## Competing interests

The authors declare no competing interests.
