## [Peer Review File · Communications Biology]

Reviewers' comments:

Reviewer #1 (Remarks to the Author):

TRAPP-like proteins and small Ras-like GTPases are known to be one of the defining features of eukaryotes. These were identified earlier in the paper reporting the superphylum of Asgard archaea. In this manuscript, the authors propose a structure-based approach for confirming the presence of dimeric TRAPP and Regulator complexes, and establish the small Ras-like GTPase fold.

The final conclusion in the abstract about membrane regulating systems is misplaced as the authors do not provide any evidence for the presence of membrane bound compartments. This conclusion is same as that in the earlier papers reporting Asgard archaea genomics and evolution. On the contrary, the data in the manuscript appears to suggest that the membrane binding function through geranyl-geranylation, as it is present in Rab GTPases, is not conserved. Additionally, the authors demonstrate that the RB7 domains also do not have a direct membrane binding function.

The data presented, though technically sound, does not provide a major conceptual advance towards understanding eukaryotic evolution, beyond what has been interpreted based on the sequence information. The authors have not exploited the structural information to derive functional insights. Simple biochemistry to demonstrate the enzymatic activity, if it exists, in the absence of a GAP, will be interesting. The point on whether GEFs are necessary, given that exchange of GDP occurs readily during crystallization also has not been explored further.

One of the major drawbacks of the manuscript is that the structural does not provide any novel functional insight beyond what has been predicted earlier from sequence analysis.

Other major concerns:

1. The main data in the manuscript is the structural characterization of the predicted small Ras-like GTPase and RB domain dimers. The rest of the structures discussed are based on AlphaFold predictions. However, it will be interesting to see if the GTPases are active, and to identify a coupled pair of GTPase and its regulator as a TRAPP/RB7/Longin domain domain.

2. Comments on the conservation of potential active site residues should be included, and a mutational study demonstrating their relevance can include interesting facts on whether a GAP is required for the GTPase. Though the RMSD and sequence conservation appears to suggest that Rab GTPase is the closest homologue, the alignment shows that the active site residues of the TtMgIA is conserved in the Thor-Rab too, as opposed to the absence of an arginine catalytic residue in Rabs. Hence, is the Asgaard archaeal GTPase a system which does not require a GAP? Or is a GAP required to orient the catalytic residues? This point might be explored in the context of the structural information of the GTP-gammaS bound active site, and further supported by GTPase activity measurements of the wild type protein and active site mutants, if the wild type protein shows activity.

3. Elaborate upon how crystal packing could result in a packing that allows nucleotide exchange. There is no figure which qualifies this statement. Small Ras-like GTPases, both prokaryotic and eukaryotic, have a tendency to bind strongly to GDP and hence require GEFs for their activation. Based on the comment on crystal packing and nucleotide exchange, is there any hypothesis on potential GEF mechanism for the Thor-Rab?

4. How were the RB and longin fold proteins chosen for characterization? There is no mention of an associated GTPase domain as coupled sequences with an RB or longin in the Asgaard archaea. In prokaryotic Ras-like GTPases, RB fold proteins often occur as a protein present within the operon. Characterization of a pair of GTPase and the coupled RB fold would have provided an interesting aspect of evolution of function in the family of proteins. Else, the characterization of GTPase and

longin/RB domains appear disconnected.

5. Another interesting aspect that could have been explored further is comparison between active site residues of Rag GTPases and the GTPase-longin domain-fusion proteins. The occurrence of RB fold, longin and profilin branches in the Asgaard archaea is an interesting observation, however this has been reported earlier too. Additional information beyond fold such as presence of extended domains, potential conservation of active site residues involved in GEF activity, etc. might be interesting insights.

Minor points:

Abstract: line 7: eukaryotic-like proteins (include 's')

Labeling the residues in the nucleotide binding pocket (shown in Supplementary Figure 2) or another new figure will be helpful in visualising potential active site residues.

Reviewer #2 (Remarks to the Author):

This collection of structural data on small GTPases from Asgard archaea will be of interest to specialists in the evolution of small GTPase regulatory systems. The structures are reported at high resolution. Various improvements are needed before the manuscript can be recommended for publication.

1. Use section headings throughout Results/Main.

2. The PDB validation reports state "resolution unknown", which does not match the manuscript, and makes it hard to evaluate if the stereochemical quality is up to expectations for the attained resolutions.

3. In the third paragraph, note that roadblock and longin domain proteins can have various regulatory and structural functions and many are not GEFs.

4. In the third paragraph, include citations for the statement that longin domain proteins can act as both GAPs and GEFs.

5. In the fourth paragraph of the Main section, Ragulator is mentioned to function as a GEF for the Rag GTPases. The Rags have intrinsic nucleotide exchange activity. There are no established Rag GEFs and it is not clear that GEFs are needed. While at one point Ragulator had been postulated to be a GEF, this was shown by Lawrence et al. 2019 Science not to be the case. Ragulator is a platform for membrane recruitment and scaffolding Rag GTPases but it is not a GEF.

6. In the paragraph discussing the Ragulator complex, the subunits are referred to as LAMPTOR 1-5. These should be LAMTOR 1-5 instead.

7. Rag GTPases can form more heterodimers than just A/C and B/D (ex. A/D or B/C as well) This should be updated in the text where it is mentioned: "Ragulator recruits heterodimer Rag GTPases (A/C or B/D)"

8. The Rag GTPases recruit mTORC1 to the lysosomal membrane in the RagAGTP:RagCGDP state.

Thor-Rab section

9. In Supplemental Figure 1 use letters for each structure so different parts of the figure can be referenced. Use different coloring for GTP and GDP bound states. Then for the superimposition, use colors consistent with the the rest of the figure to denote the GTP and GDP bound conformations. Label switch I and II.

10. Why was Thor-Rab crystallized in the absence of Mg²⁺? Was the intention to destabilize all nucleotides in the binding site?

11. Why is there a mixed population of nucleotides in the crystal? Is GDP the preferred binding state or was there GDP present in the sample previously or supplemented in the buffer during purification?

It would be better to prepare samples of defined nucleotide state and determine these structures.

12. The variability in the switch regions in the two molecules in the asymmetric unit is a little odd. For example, in supplement Figure 1, for the crystal with both GDP bound, switch II is ordered in one molecule and disordered in the other. Possibly the ordered copy is in a crystal contact? Please comment. For the GTPyS bound state, switch II is ordered in the first molecule and disordered in the second, which is odd, since in most GTPases switch II is ordered in the GTP state.

13. For figure 1, colors for Thor-Rab should be consistent across panels A, B and C for clarity. Label where the core, switch I and II are located on the structure in panels B and C.

14. With the absence of the C-terminal cysteine residues, Thor-Rab might be modified differently than eukaryotic Rags for insertion into membranes. Couldn't it also be possible that Thor-Rab is not inserted into membranes at all and has a different function?

RB section

15. Reference to figure 2a-c for the below sentence should be sufficient instead of 2a-f

"Both proteins are homodimers formed from protomers comprised of 5-stranded β -sheets sandwiched between a single α -helix (formed between strands 2 and 3) and a pair of α -helices (formed from the N- and C-termini) (Fig. 2a-f)."

16. Move the text below comparing the surface charges between Thor-RB and MKD1-RB with eukaryotic proteins to the next paragraph that is discussing similarities and differences between these proteins.

"Despite the similarity in structures, Thor-RB and MKD1-RB display different surface charge distributions (Fig. 2d,e). MKD1-RB has a basic patch on the same face as observed for MglB, which in the case of MglB, is used to recruit the MglA/MglB complex to membranes (Fig. 2e, i, k)²⁶. Thus, we hypothesize that MKD1-RB may be a membrane-interacting module, whereas Thor-RB, which lacks the basic patch, may be a scaffold protein for protein:protein interactions."

RBL7 section

17. The phrase "refined against at 1.83A data", delete "at".

18. Show an overlay or side by side view of the RB and RBL7 proteins referenced in the below sentence to show the difference in the structures (i.e loss of terminal helices). This overlay could be added to a main text figure or a supplemental figure and referenced instead of Fig. 2a,b at the end of the below quote. Annotate the missing terminal pair of helices.

19. In Figure 3, keep the orientations the same for the three proteins in the comparison.

20. In the Figure 3 legend, refer to panel (C) and (D) separately from each other. Same comment for (E,F) portion of the figure legend.

21. In the section indicated below, it is unclear how this ties back to the original point in the paragraph. For clarification, it may be helpful to discuss its structural similarities and differences to the Odin RBL7 protein as shown in supplemental figure 5 that is referenced in the text. Supplementary Fig. 5B is referenced twice in the sentence for different protein structures. Clarify which is the correct reference to the figure.

"We were not successful in solving the structure of a longin domain, however many such sequences are predicted in the Asgard genomes^{17,18}. We used AF2 to explore the structure of a longin domain protein from MKD1 (Supplementary Fig. 5b), which was predicted to have similar fold to eukaryotic longin domains (Supplementary Fig. 5B)"

22. In the sentence below, the word choice for substituted is confusing. The helices are not replaced at the same location but rather they one helix in the protein is lost and another is gained at a different location. Rephrase to clarify.

“longin domains have the N-terminal helix substituted by an additional C-terminal helix (Fig. 4a-c).”

23. In the Figure 4 legend, indicate what the different colors in the structure cartoon represent in A-D with relation to the structural features. Additionally, are these cartoons just for archaea or eukaryotic proteins discussed as well? Clarify this point in the figure legend or text.

24. In Figure 4F, highlight (arrow or other means) where the dimerization interface as well as the actin binding interface discussed in the text occurs.

“This region is responsible for dimerization in RB proteins and in actin binding in profilin”

25. Point to specific structural details from your structures or other structures that would support this point. It is unclear from the data in Figure 4, which is discussed previously in this paragraph, how this claim was made.

“It is likely that differentiation in these proteins occurred in the ancestors of Asgard archaea and the expansion in numbers of each fold continued in different Asgard lineages.”

TRAPPC3 section

26. In the TRAPPC3 section two different proteins are mentioned in the beginning as being purified and crystallized. Later it becomes unclear which protein structure is being referred to since they are both labelled Thor-TRAPPC3. Use two consistent but distinct names for them. One possibility is to use a superscript such as Thor-TRAPPC3^{SMTZ1} and Thor-TRAPPC3^{AB25}. Similarly, in Figure 5, it is unclear which Thor-TRAPPC3 protein is being visualized in panel A and E.

27. “Thor-TRAPPC3 is structurally more distant to the bacterial V4R domains, such as the PoxR homodimer” While this statement is true, the global architecture of the GTPases is very similar. State this or indicate in the comparison figure the differences you are highlighting.

28. A main point in the text is the presence of a cavity in Thor-TRAPPC3. In Figure 5 panels E, F, and G, it would be helpful to have the charge of the cavities themselves and a surface representation instead of the charge around the outside of the structure.

Reviewer #3 (Remarks to the Author):

This manuscript structurally characterizes several membrane trafficking proteins in Archea using both X-Ray crystallography and alphafold modelling. The authors compare these archaeal structures to structures of similar proteins in bacteria and eukaryotes, to investigate the evolution of membrane trafficking proteins. This work does add to our understanding of archaeal-eukaryotic membrane trafficking evolution, and I enjoyed reading this well written manuscript, however I do have some concerns before I would recommend the manuscript for publication.

Major Concerns:

- 1) Mixed occupancy for the Thor-Rab is likely due to the lack of efficient nucleotide loading. I would recommend in the future treating the Rabs with phosphatase and EDTA prior to the addition of nucleotide to ensure that loading is complete. It is intriguing that there are no differences in the switch I and II conformations in between GDP and GTP bound states, can the authors comment on the symmetry of the crystal packing? It would be good to add an additional figure with the crystal packing of the asymmetric unit, as sometimes crystal packing can alter the switch conformations.
- 2) Alphafold predictions are extremely useful, but currently as presented there is no way to validate

the likelihood of these folds, as no PAE (predicted aligned error) plots are shown. There are nice PAE tutorials on the alphafold website (<https://alphafold.ebi.ac.uk/entry/Q9Y223>). The structures should be colored by the plddt score, to indicate to readers the confidence in each region with a clear legend. There are also no methods for how these searches were conducted. I think it is important that these be added to the manuscript, as without this data the alphafold search is not reliable.

3) The authors calculate a phylogenetic tree from the structure-based alignment, and I believe a methods section should be added for how the alignments and phylogenetic analyses were completed.

4) The discovery that Thor-TRAPPC3 shows similar folding to TRAPPC3/C6 complex is intriguing, have the reviewers seen any evidence for TRAPPC4 like (the component of TRAPP II/III that has GEF activity), or any other TRAPP proteins in Archea? It would be interesting to see if this early TRAPP protein/complex does not have GEF activity and acts only as a scaffold.

Minor Concerns:

1) Figure 1b the colors are very difficult to distinguish. I would recommend changing these to make it more clear.

2) Methods section for protein purification, page 12, you mention using "protease inhibitor cocktail free edta". Do you mean to say edta-free?

3) In the abstract, the authors say "...relatedness of these eukaryotic-like protein" – should be changed to "proteins".

4) Thor-rab results section "authenticity of Asgard Rab-type small GTPases" – should this say Rab-like not Rab-type?

5) The authors comment that there are no c-terminal cysteine residues in Thor-Rab, but are there any genes involved in prenylation present in the archeal genome (Rab GGT, GDI, REP)? With how different membranes are between eukarya and archea I wouldn't necessarily expect it to be present, but if the reviewers have looked it would be nice to comment on this.

Reviewer #1 (Remarks to the Author):

TRAPP-like proteins and small Ras-like GTPases are known to be one of the defining features of eukaryotes. These were identified earlier in the paper reporting the superphylum of Asgard archaea. In this manuscript, the authors propose a structure-based approach for confirming the presence of dimeric TRAPP and Ragulator complexes, and establish the small Ras-like GTPase fold. The final conclusion in the abstract about membrane regulating systems is misplaced as the authors do not provide any evidence for the presence of membrane bound compartments. This conclusion is same as that in the earlier papers reporting Asgard archaea genomics and evolution. On the contrary, the data in the manuscript appears to suggest that the membrane binding function through geranyl-geranylation, as it is present in Rab GTPases, is not conserved. Additionally, the authors demonstrate that the RB7 domains also do not have a direct membrane binding function.

We thank the reviewer for the comment and agree that the emphasis was misleading. This sentence is now changed to: “We conclude that the emergence of these protein architectures predated eukaryogenesis, however further adaptations occurred in proto-eukaryotes to allow these proteins to regulate distinct internal membranes.”

The data presented, though technically sound, does not provide a major conceptual advance towards understanding eukaryotic evolution, beyond what has been interpreted based on the sequence information. The authors have not exploited the structural information to derive functional insights. Simple biochemistry to demonstrate the enzymatic activity, if it exists, in the absence of a GAP, will be interesting. The point on whether GEFs are necessary, given that exchange of GDP occurs readily during crystallization also has not been explored further.

We have added a phosphate release assay (Fig. 1g,h), which shows that the innate GTPase activity is similar to eukaryotic Rabs. We comment: “This indicates that the unassisted GTPase activity of SMTZ1-45 Thor-Rab is similar to eukaryotic Rab proteins, implying that a GAP and/or GEF may be needed to accelerate GTP hydrolysis for signaling.”

The potential GAP/GEFs are likely to be the RB and longin proteins. Since these are dimers, and possibly heterodimers, with close to 100 sequences in a single genome, we have no way at present to match the GTPase to its potential regulators as they do not form operons. Such analysis is beyond the current manuscript.

We have further analysed the crystal packing to reveal the cause of nucleotide exchange (New Fig. 2).

One of the major drawbacks of the manuscript is that the structural does not provide any novel functional insight beyond what has been predicted earlier from sequence analysis. From sequences one can speculate/predict the potential structures and functions of proteins. Here, we provide experimental data to demonstrate: 1) that Thor-Rab structure has a genuine Rab core and that it hydrolyses GTP slowly; 2) the RB proteins form dimers which have a variety of surface features that suggest a spectrum of functions; 3) that TRAPPC3 proteins form dimers with cavities, and identify zinc binding sites that place these proteins at the interface between ligand-binding proteins and eukaryotic TRAPPC3/C6; 4) From the zinc-binding sites we were able to identify other homologs not found by regular sequence searches. These insights could not be gained from sequences alone. We hope the

reviewer can accept these insights as being interesting, and appreciate that our structural work places more certainty on the previous predictions from sequence analyses.

Other major concerns:

1. The main data in the manuscript is the structural characterization of the predicted small Ras-like GTPase and RB domain dimers. The rest of the structures discussed are based on AlphaFold predictions.

We also report the first crystal structures of prokaryotic TRAPPC3-like proteins.

However, it will be interesting to see if the GTPases are active, and to identify a coupled pair of GTPase and its regulator as a TRAPP/RB7/Longin domain domain.

We agree. As stated above, we have added a phosphate release assay (Fig. 1g,h), which shows that the innate GTPase activity is similar to eukaryotic Rabs.

Identifying the specific GTPase regulators is complex. RB/longin are dimers, and possibly heterodimers, with close to 100 sequences in a single genome, we have no way at present to match the GTPases to their potential regulators as they do not form operons. This compounded by there being 74+ GTPases in MKD1. We feel that discovering a coupled pair is beyond the scope of the present manuscript.

2. Comments on the conservation of potential active site residues should be included, and a mutational study demonstrating their relevance can include interesting facts on whether a GAP is required for the GTPase. Though the RMSD and sequence conservation appears to suggest that Rab GTPase is the closest homologue, the alignment shows that the active site residues of the TtMglA is conserved in the Thor-Rab too, as opposed to the absence of an arginine catalytic residue in Rabs. Hence, is the Asgaard archaeal GTPase a system which does not require a GAP? Or is a GAP required to orient the catalytic residues? This point might be explored in the context of the structural information of the GTP-gammaS bound active site, and further supported by GTPase activity measurements of the wild type protein and active site mutants, if the wild type protein shows activity.

These are excellent points. We now further analyse the sequences. We find that Thor-Rab is an oddity in having an arginine in the active site. We have added sequence alignments in Sup. Figs 3, 5, 6, 7. These data predict that only the MglA-like sequences (Sup. Fig. 7) consistently have the catalytic arginine and extended Switch I loop, the others likely require GAPs. We comment: "Thor-Rab has an arginine residue (Arg37) in an equivalent position in the sequence alignment to MglA Arg53 in Switch I (red triangle, Fig. 1d and Supplementary Fig. 2g). This residue in MglA has been predicted to act as an intrinsic "Arg finger" in stabilizing the GTP γ -phosphate during hydrolysis²⁵. Other Rab paralogs from the Thor SMTZ1-45 genome do not have an arginine residue in this position (Supplementary Fig. 3). Thus, the MglA mechanism of γ -phosphate self-stabilization by arginine is not a common feature in Thor Rabs, implying that they may require GAPs to enhance hydrolysis."

3. Elaborate upon how crystal packing could result in a packing that allows nucleotide exchange. There is no figure which qualifies this statement. Small Ras-like GTPases, both prokaryotic and eukaryotic, have a tendency to bind strongly to GDP and hence require GEFs for their activation. Based on the comment on crystal packing and nucleotide exchange, is there any hypothesis on potential GEF mechanism for the Thor-Rab?

We have analysed the crystal packing to reveal the cause of nucleotide exchange (Fig. 2).

4. How were the RB and longin fold proteins chosen for characterization? There is no mention of an associated GTPase domain as coupled sequences with an RB or longin in the Asgaard archaea. In prokaryotic Ras-like GTPases, RB fold proteins often occur as a protein present within the operon. Characterization of a pair of GTPase and the coupled RB fold would have provided an interesting aspect of evolution of function in the family of proteins. Else, the characterization of GTPase and longin/RB domains appear disconnected.

In general, the GTPase domain proteins do not occur in operons and when they do rarely with RB or longin domain proteins. In the MK-D1 genome we found two MglA-like GTPases in operons with RB domain proteins, and one each of a RB or longin domain protein in an operon with other GTPases. However, these last two cases involve larger proteins, which may not good models for general GTPase regulation. We conclude that the use of operons is not a major method of regulation of GTPase/regulator co-expression.

We comment in the RB section:

“These proteins were chosen based sequence homology to eukaryotic RB domains.”

And comment in the discussion:

In the discussion we comment “However, MK-D1 small GTPase genes do not generally exist within operons with RB, longin, or TRAPPC3 genes. This indicates that the use of operons is not a major method of regulation of GTPase/regulator co-expression, and perhaps suggests that transcription factor specific regulation may be used. This makes matching of individual small GTPases to potential GAPs or GEFs difficult to predict from the genome lay out.”

5. Another interesting aspect that could have been explored further is comparison between active site residues of Rag GTPases and the GTPase-longin domain-fusion proteins. The occurrence of RB fold, longin and profilin branches in the Asgaard archaea is an interesting observation, however this has been reported earlier too. Additional information beyond fold such as presence of extended domains, potential conservation of active site residues involved in GEF activity, etc. might be interesting insights.

We have now added sequence alignments that are grouped by GTPase architectures (Supplementary Fig. 5-7). We comment on the absence of the activating arginine residue from all the GTPases except for the MglA-like GTPases from the MK-D1 genome in the figure legends. We also comment on an insertion in some of the RB GTPases Switch II in the main text. “Interestingly, an insert was observed in Switch II from GTPase-RB architectures but not in RB-GTPase architectures, indicating potential differences in nucleotide regulation between these designs (Supplementary Fig. 6a).”

Minor points:

Abstract: line 7: eukaryotic-like proteins (include 's')

Changed

Labeling the residues in the nucleotide binding pocket (shown in Supplementary Figure 2) or another new figure will be helpful in visualising potential active site residues.

Selected residues are now labelled.

Reviewer #2 (Remarks to the Author):

This collection of structural data on small GTPases from Asgard archaea will be of interest to specialists in the evolution of small GTPase regulatory systems. The structures are reported at high resolution. Various improvements are needed before the manuscript can be recommended for publication.

Thank you for the positive assessment.

1. Use section headings throughout Results/Main.

We have added more section headings.

2. The PDB validation reports state "resolution unknown", which does not match the manuscript, and makes it hard to evaluate if the stereochemical quality is up to expectations for the attained resolutions.

We submit the updated reports.

3. In the third paragraph, note that roadblock and longin domain proteins can have various regulatory and structural functions and many are not GEFs.

We now comment: "Some RB/longin domain proteins have roles that are not involved in small GTPase regulation, for instance longin domains are found in SNARE proteins, however these other functions often involve membrane modulation. A second system is the Ragulator-Rag complex which is involved in localizing the TORC1 metabolic sensing complex on the lysosome."

4. In the third paragraph, include citations for the statement that longin domain proteins can act as both GAPs and GEFs.

Now included.

5. In the fourth paragraph of the Main section, Ragulator is mentioned to function as a GEF for the Rag GTPases. The Rags have intrinsic nucleotide exchange activity. There are no established Rag GEFs and it is not clear that GEFs are needed. While at one point Ragulator had been postulated to be a GEF, this was shown by Lawrence et al. 2019 Science not to be the case. Ragulator is a platform for membrane recruitment and scaffolding Rag GTPases but it is not a GEF.

Thank you for this clarification. We have changed the text to:

"The Ragulator complex acts as a scaffold to recruit heterodimer Rag GTPases (A/C, A/D, B/C or B/D), and the longin domain-containing Foliculin/Foliculin interacting protein (FLCN:FNIP) complex, to the lysosome. Nutrient stimulation stimulates RagA and the FLCN:FNIP complex to promote exchange of the nucleotides from GDP_{RagA}:RagCGTP to GTP_{RagA}:RagCGDP, to engage TORC1 in regulating lysosome activity, biogenesis and positioning."

6. In the paragraph discussing the Ragulator complex, the subunits are referred to as LAMPTOR 1-5. These should be LAMTOR 1-5 instead.

Thank you – this is now corrected.

7. Rag GTPases can form more heterodimers than just A/C and B/D (ex. A/D or B/C as well)
This should be updated in the text where it is mentioned: "Regulator recruits heterodimer Rag GTPases (A/C or B/D)"

This now corrected.

8. The Rag GTPases recruit mTORC1 to the lysosomal membrane in the RagAGTP:RagCGDP state.

This is now corrected.

Thor-Rab section

9. In Supplemental Figure 1 use letters for each structure so different parts of the figure can be referenced. Use different coloring for GTP and GDP bound states. Then for the superimposition, use colors consistent with the rest of the figure to denote the GTP and GDP bound conformations. Label switch I and II.

Implemented as suggested.

10. Why was Thor-Rab crystallized in the absence of Mg²⁺? Was the intention to destabilize all nucleotides in the binding site?

The intention was to purify the apo form and then add back the nucleotides and magnesium. However the protein still purified with GDP.

11. Why is there a mixed population of nucleotides in the crystal? Is GDP the preferred binding state or was there GDP present in the sample previously or supplemented in the buffer during purification? It would be better to prepare samples of defined nucleotide state and determine these structures.

Thor-Rab shows slow nucleotide hydrolysis, which is fast compared to the time for crystallization. This is seen by comparing the GTP-γS soak vs the co-crystallization. In the soak we were able to replace 100% of the nucleotide due to the short time period (10 min). The GTP-γS co-crystallization can be considered a defined GTP-γS-bound state, since in the soak GTP-γS replaces the GDP. However in GTP-γS co-crystallization, we observe GDP, which we believe is due to hydrolysis in the crystal or during crystallization.

12. The variability in the switch regions in the two molecules in the asymmetric unit is a little odd. For example, in supplement Figure 1, for the crystal with both GDP bound, switch II is ordered in one molecule and disordered in the other. Possibly the ordered copy is in a crystal contact? Please comment. For the GTPγS bound state, switch II is ordered in the first molecule and disordered in the second, which is odd, since in most GTPases switch II is ordered in the GTP state.

We now analyze the crystal contacts and compare more thoroughly to other structures in Fig. 3 and 4, and comment in the legends.

13. For figure 1, colors for Thor-Rab should be consistent across panels A, B and C for clarity. Label where the core, switch I and II are located on the structure in panels B and C.

Implemented as suggested.

14. *With the absence of the C-terminal cysteine residues, Thor-Rab might be modified differently than eukaryotic Rags for insertion into membranes. Couldn't it also be possible that Thor-Rab is not inserted into membranes at all and has a different function?*

We don't know the function of ThorRab. However, since the imaged Asgard archaea have no internal membranes, any membrane-binding Rabs will not encounter the same issues as eukaryotic Rabs, which need to distinguish between membranes. We now discuss this. "These cysteine residues are absent from Thor-Rab and the C-terminus is significantly truncated relative to eukaryotic Rabs. This indicates that Thor-Rab is not modified for insertion into membranes in the same manner as eukaryotic Rabs. As far as we know, Asgard archaea only have one membrane, the cell membrane^{27,28}. Thus, the Rab cysteine-containing C-terminal extension likely arose in proto-eukaryotes in conjunction with the acquisition and distinction in internal membranes."

RB section

15. *Reference to figure 2a-c for the below sentence should be sufficient instead of 2a-f*

"Both proteins are homodimers formed from protomers comprised of 5-stranded β -sheets sandwiched between a single α -helix (formed between strands 2 and 3) and a pair of α -helices (formed from the N- and C-termini) (Fig. 2a-f)."

Implemented as suggested.

16. *Move the text below comparing the surface charges between Thor-RB and MKD1-RB with eukaryotic proteins to the next paragraph that is discussing similarities and differences between these proteins.*

"Despite the similarity in structures, Thor-RB and MKD1-RB display different surface charge distributions (Fig. 2d,e). MKD1-RB has a basic patch on the same face as observed for MglB, which in the case of MglB, is used to recruit the MglA/MglB complex to membranes (Fig. 2e, i, k)²⁶. Thus, we hypothesize that MKD1-RB may be a membrane-interacting module, whereas Thor-RB, which lacks the basic patch, may be a scaffold protein for protein:protein interactions."

Implemented as suggested.

RBLC7 section

17. *The phrase "refined against at 1.83A data", delete "at".*

The typo is corrected.

18. *Show an overlay or side by side view of the RB and RBLC7 proteins referenced in the below sentence to show the difference in the structures (i.e loss of terminal helices). This overlay could be added to a main text figure or a supplemental figure and referenced instead of Fig. 2a,b at the end of the below quote. Annotate the missing terminal pair of helices.*

The requested figure has been added. Fig. 6.

19. In Figure 3, keep the orientations the same for the three proteins in the comparison. The orientations are superimpositions, which are dominated via the B-sheets.

20. In the Figure 3 legend, refer to panel (C) and (D) separately from each other. Same comment for (E,F) portion of the figure legend.

Implemented as suggested.

21. In the section indicated below, it is unclear how this ties back to the original point in the paragraph. For clarification, it may be helpful to discuss its structural similarities and differences to the Odin RBLC7 protein as shown in supplemental figure 5 that is referenced in the text. Supplementary Fig. 5B is referenced twice in the sentence for different protein structures. Clarify which is the correct reference to the figure.

“We were not successful in solving the structure of a longin domain, however many such sequences are predicted in the Asgard genomes^{17,18}. We used AF2 to explore the structure of a longin domain protein from MKD1 (Supplementary Fig. 5b), which was predicted to have similar fold to eukaryotic longin domains (Supplementary Fig. 5B)”

The reference has been clarified and the section has been separated from the RBLC7 section.

22. In the sentence below, the word choice for substituted is confusing. The helices are not replaced at the same location but rather they one helix in the protein is lost and another is gained at a different location. Rephrase to clarify.

“longin domains have the N-terminal helix substituted by an additional C-terminal helix (Fig. 4a-c).”

We have clarified: “RBLC7 has lost the C-terminal helix relative to RB, while the longin domain has lost the N-terminal helix and instead placed an additional C-terminal helix at the same location”.

23. In the Figure 4 legend, indicate what the different colors in the structure cartoon represent in A-D with relation to the structural features. Additionally, are these cartoons just for archaea or eukaryotic proteins discussed as well? Clarify this point in the figure legend or text.

Updated as suggested.

24. In Figure 4F, highlight (arrow or other means) where the dimerization interface as well as the actin binding interface discussed in the text occurs.

“This region is responsible for dimerization in RB proteins and in actin binding in profilin”

Highlighting arrows are now included.

25. Point to specific structural details from your structures or other structures that would support this point. It is unclear from the data in Figure 4, which is discussed previously in this paragraph, how this claim was made.

“It is likely that differentiation in these proteins occurred in the ancestors of Asgard archaea and the expansion in numbers of each fold continued in different Asgard lineages.”

The sentence is based on the numbers of each fold found in the different lineages. We have clarified this point: *“It is likely that differentiation in these proteins occurred in the ancestors of Asgard archaea, since these proteins are found in all Asgard archaea phyla, and expansion in numbers of each fold continued in the subsequent Asgard lineages leading to the variations between lineages, as documented for profilins.”*

TRAPPC3 section

26. *In the TRAPPC3 section two different proteins are mentioned in the beginning as being purified and crystallized. Later it becomes unclear which protein structure is being referred to since they are both labelled Thor-TRAPPC3. Use two consistent but distinct names for them. One possibility is to use a superscript such as Thor-TRAPPC3^{SMTZ1} and Thor-TRAPPC3^{AB25}. Similarly, in Figure 5, it is unclear which Thor-TRAPPC3 protein is being visualized in panel A and E.*

The TRAPPC3 proteins are now clearly indicated, generally by **SMTZ1 Thor-TRAPPC3** and **AB25 Thor-TRAPPC3**.

27. *“Thor-TRAPPC3 is structurally more distant to the bacterial V4R domains, such as the PoxR homodimer” While this statement is true, the global architecture of the GTPases is very similar. State this or indicate in the comparison figure the differences you are highlighting.*

This has been clarified: *“Bacterial V4R domains, such as the PoxR homodimer, are also structurally similar to SMTZ1-45 Thor-TRAPPC3 (Fig. 5b,d and Supplementary Table 7, 130 matching residues, RMSD of 3.3 Å), albeit more distant than eukaryotic TRAPPC3/C6.”*

28. *A main point in the text is the presence of a cavity in Thor-TRAPPC3. In Figure 5 panels E, F, and G, it would be helpful to have the charge of the cavities themselves and a surface representation instead of the charge around the outside of the structure.*

The charged cavity is now included in Fig. 8.

Reviewer #3 (Remarks to the Author):

This manuscript structurally characterizes several membrane trafficking proteins in Archaea using both X-Ray crystallography and alphaFold modelling. The authors compare these archaeal structures to structures of similar proteins in bacteria and eukaryotes, to investigate the evolution of membrane trafficking proteins. This work does add to our understanding of archaeal-eukaryotic membrane trafficking evolution, and I enjoyed reading this well written manuscript, however I do have some concerns before I would recommend the manuscript for publication.

Thank you for the positive comments.

Major Concerns:

1) *Mixed occupancy for the Thor-Rab is likely due to the lack of efficient nucleotide loading. I would recommend in the future treating the Rabs with phosphatase and EDTA prior to the addition of nucleotide to ensure that loading is complete. It is intriguing that there are no*

differences in the switch I and II conformations in between GDP and GTP bound states, can the authors comment on the symmetry of the crystal packing? It would be good to add an additional figure with the crystal packing of the asymmetric unit, as sometimes crystal packing can alter the switch conformations.

We have analysed the crystal packing to reveal the cause of the loop conformations (Fig. 2). We have also added a phosphate release assay (Fig. 1g,h), which shows that the innate GTPase activity is similar to eukaryotic Rabs. Thus, Thor-Rab shows slow nucleotide hydrolysis, which is fast compared to the time for crystallization. This is seen by comparing the GTP- γ S soak vs the co-crystallization. In the soak we were able to replace 100% of the nucleotide due to the short time period (10 min). However in GTP- γ S co-crystallization, we observe GDP, which we believe is due to hydrolysis in the crystal or during crystallization.

2) Alphafold predictions are extremely useful, but currently as presented there is no way to validate the likelihood of these folds, as no PAE (predicted aligned error) plots are shown. There are nice PAE tutorials on the alphafold website (<https://alphafold.ebi.ac.uk/entry/Q9Y223>). The structures should be colored by the plddt score, to indicate to readers the confidence in each region with a clear legend. There are also no methods for how these searches were conducted. I think it is important that these be added to the manuscript, as without this data the alphafold search is not reliable.

These have now been added. Please see Fig. S8, S9 and S10. Comments on the PAE plots with regards to the confidence in the predictions are given in the legends.

3) The authors calculate a phylogenetic tree from the structure-based alignment, and I believe a methods section should be added for how the alignments and phylogenetic analyses were completed.

We now state: "Sequence alignment and phylogenetic analysis were carried out in MAAFT⁴² and structural comparisons in Dali²⁹ and PDBeFold⁴³."

4) The discovery that Thor-TRAPPC3 shows similar folding to TRAPPC3/C6 complex is intriguing, have the reviewers seen any evidence for TRAPPC4 like (the component of TRAPP II/III that has GEF activity), or any other TRAPP proteins in Archea? It would be interesting to see if this early TRAPP protein/complex does not have GEF activity and acts only as a scaffold.

This is a great question, but difficult experimentally. We have no way at present to match the GTPases to their potential regulators as they do not exist in operons. This compounded by there being 74+ GTPases in MKD1, with similarly large numbers in the Thor partial genomes. We hope to address this in the future.

Minor Concerns:

1) Figure 1b the colors are very difficult to distinguish. I would recommend changing these to make it more clear.

We have changed the colors in line with this comment and those from one of the other reviewers.

2) Methods section for protein purification, page 12, you mention using "protease inhibitor cocktail free edta". Do you mean to say edta-free?

Corrected.

3) In the abstract, the authors say “...relatedness of these eukaryotic-like protein” – should be changed to “proteins”.

Corrected.

4) Thor-rab results section “authenticity of Asgard Rab-type small GTPases” – should this say Rab-like not Rab-type?

Corrected.

5) The authors comment that there are no c-terminal cysteine residues in Thor-Rab, but are there any genes involved in prenylation present in the archeal genome (Rab GGT, GDI, REP)? With how different membranes are between eukarya and archea I wouldn't necessarily expect it to be present, but if the reviewers have looked it would be nice to comment on this.

We now comment:

“Similarly, we could not find homologs for the eukaryotic Rab Escort Protein 1 and Rab geranylgeranyltransferase subunits in sequence databases using BLAST.”

REVIEWERS' COMMENTS:

Reviewer #1 (Remarks to the Author):

The manuscript reporting the structural analysis of Rab GTPases and RB/longin domains from Archaea has been resubmitted after revision.

The authors have revised the manuscript satisfactorily. Though it would have been interesting to have a functional correlation between the Rab GTPases and the longin domains discussed in the manuscript, the authors have now explained their difficulties in identifying a GTPase and its corresponding functionally interacting pair.

The manuscript may be accepted in the current form.

I have mentioned a few minor concerns below which the authors can check and rectify if necessary.

Figure 2: Does the density really correspond to Mg²⁺? What is the coordinating geometry/distances?

Please carry out for a validity check of interpretation of Mg²⁺ density. Long coordination distances might imply the presence of a water molecule rather than Mg²⁺?

Also the table 1a mention site A and Site B for Mg²⁺. Are these two sites relevant? A shift in the Mg²⁺ might be accompanied by shift in the phosphate position as well?

Minor corrections:

Lines 109 - 111 _ LAMPTOR to LAMTOR.

Supplementary page 16: Correct RSMD to RMSD

Reviewer #2 (Remarks to the Author):

Most comments were addressed well, but a few comments were not addressed fully. These are both in paragraph 4 of the manuscript.

1. In the paragraph discussing the Ragulator complex, the subunits are referred to as LAMPTOR 1-5. These should be LAMTOR 1-5 instead.

2. Rag GTPases can form more heterodimers than just A/C and B/D (ex. A/D or B/C as well) This should be updated in the text where is is mentioned: "Ragulator recruits heterodimer Rag GTPases (A/C or B/D)"

Reviewer #3 (Remarks to the Author):

Thank you for addressing my concerns. The additional new figures and comments strengthen the article. I have no major concerns.

Reviewer #1 (Remarks to the Author):

Figure 2: Does the density really correspond to Mg^{2+} ? What is the coordinating geometry/distances?

Please carry out for a validity check of interpretation of Mg^{2+} density. Long coordination distances might imply the presence of a water molecule rather than Mg^{2+} ?

The average coordination lengths are now given in the figure legend. These are appropriate for Mg^{2+} , and inappropriate for water.

Also the table 1a mention site A and Site B for Mg^{2+} . Are these two sites relevant? A shift in the Mg^{2+} might be accompanied by shift in the phosphate position as well?

Our wording was misleading. There is only one site per protein chain. We have changed the wording to "Occupancy of Mg^{2+} site in molecule A" etc

Minor corrections:

Lines 109 - 111 _ LAMPTOR to LAMTOR.

Supplementary page 16: Correct RSMD to RMSD

Corrected

Reviewer #2 (Remarks to the Author):

Most comments were addressed well, but a few comments were not addressed fully. These are both in paragraph 4 of the manuscript.

1. In the paragraph discussing the Ragulator complex, the subunits are referred to as LAMPTOR 1-5. These should be LAMTOR 1-5 instead.

Changed.

2. Rag GTPases can form more heterodimers than just A/C and B/D (ex. A/D or B/C as well) This should be updated in the text where is is mentioned: "Ragulator recruits heterodimer Rag GTPases (A/C or B/D)"

Changed.

Best wishes,

Bob